# Evidence for European presence in the Americas in AD 1021

Margot Kuitems[1✉], Birgitta L. Wallace[2], Charles Lindsay[2], Andrea Scifo[1], Petra Doeve[3,4], Kevin Jenkins[2], Susanne Lindauer[5], Pınar Erdil[1], Paul M. Ledger[6,7], Véronique Forbes[6], Caroline Vermeeren[8], Ronny Friedrich[5] & Michael W. Dee[1✉]

Transatlantic exploration took place centuries before the crossing of Columbus. Physical evidence for early European presence in the Americas can be found in Newfoundland, Canada[1,2]. However, it has thus far not been possible to determine when this activity took place[3–5]. Here we provide evidence that the Vikings were present in Newfoundland in AD 1021. We overcome the imprecision of previous age estimates by making use of the cosmic-ray-induced upsurge in atmospheric radiocarbon concentrations in AD 993 (ref. [6]). Our new date lays down a marker for European cognisance of the Americas, and represents the first known point at which humans encircled the globe. It also provides a definitive tie point for future research into the initial consequences of transatlantic activity, such as the transference of knowledge, and the potential exchange of genetic information, biota and pathologies[7,8].

The Vikings (or Norse) were the first Europeans to cross the Atlantic[9]. However, the only confirmed Norse site in the Americas is L'Anse aux Meadows, Newfoundland[9–12] (Extended Data Figs. 1 and 2). Extensive field campaigns have been conducted at this UNESCO (United Nations Educational, Scientific, and Cultural Organization) World Heritage Site, and much knowledge has been gained about the settlement and its contemporary environment[2,13–15] (Supplementary Note 1). Evidence has also revealed that L'Anse aux Meadows was a base camp from which other locations, including regions further south, were explored[15].

The received paradigm is that the Norse settlement dates to the close of the first millennium[9]; however, the precise age of the site has never been scientifically established. Most previous estimates have been based on stylistic analysis of the architectural remains and a handful of artefacts, as well as interpretations of the Icelandic sagas, oral histories that were only written down centuries later[2,16] (Supplementary Note 2). Radiocarbon ([14]C) analysis has been attempted at the site, but has not proved especially informative[3,17,18]. More than 150 [14]C dates have been obtained, of which 55 relate to the Norse occupation[19]. However, the calibrated age ranges provided by these samples extend across and beyond the entire Viking Age (AD 793–1066) (Fig. 1 and Extended Data Fig. 3). This is in contrast with the archaeological evidence and interpretations of the sagas. The latter offer differing scenarios for the frequency and duration of Norse activity in the Americas, but both the archaeological and written records are consistent with a very brief occupation (Supplementary Note 3 and Extended Data Fig. 4). The unfavourable spread in the [14]C dates is down to the limitations of this chronometric technique in the 1960s and 1970s when most of these dates were obtained. Such impediments included far greater measurement uncertainty and restrictive sample size requirements. Furthermore,

many of these samples were subject to an unknown amount of inbuilt age. The term inbuilt age refers to the difference in time between the contextual age of the sample and the time at which the organism died (returned by [14]C analysis), which can potentially reach hundreds of years. This offset was also sometimes inappropriately incorporated into summary estimates[3].

## Cosmic radiation events as absolute time markers

In our study, we use an advanced chronometric approach to anchor Norse activity in the Americas to a precise point in time. Exact-year [14]C results can be achieved by high-precision accelerator mass spectrometry (AMS) in combination with distinct features in the atmospheric [14]C record[20–22]. Measurements on known-age (dendrochronological) tree rings show that [14]C production usually fluctuates by less than 2‰ per year[23]. However, such time series have also revealed that production of the isotope rapidly increased in the years AD; 775 and AD 993 by about 12‰ (which manifests as a decrease of about 100 [14]C yr)[24] and about 9‰ (about 70 [14]C yr)[6], respectively. These sudden increases were caused by cosmic radiation events, and appear synchronously in dendrochronological records all around the world[25–29]. By uncovering these features in tree-ring samples of unknown age, it is possible to effect precise pattern matching between such samples and reference series. In so doing, if the bark edge (or more specifically, the waney edge) is also present, it becomes possible to determine the exact felling year of the tree[20]. Moreover, it is not necessary to have [14]C dates for the outermost growth rings, because once the ring that contains the AD 993 anomaly has been detected, it simply becomes a matter of counting the number of rings to the waney edge. On the basis of the state of development of the earlywood and latewood

[1]Centre for Isotope Research, University of Groningen, Groningen, the Netherlands. [2]Parks Canada Agency, Government of Canada, Dartmouth, Nova Scotia, Canada. [3]Laboratory for Dendrochronology at BAAC, 's-Hertogenbosch, the Netherlands. [4]Cultural Heritage Agency of The Netherlands, Amersfoort, the Netherlands. [5]Curt-Engelhorn-Center Archaeometry, Mannheim, Germany. [6]Department of Archaeology, Queens College, Memorial University of Newfoundland, St Johns, Newfoundland, Canada. [7]Department of Geography, Memorial University of Newfoundland, St Johns, Newfoundland, Canada. [8]BIAX Consult, Zaandam, the Netherlands. ✉e-mail: m.kuitems@rug.nl; m.w.dee@rug.nl

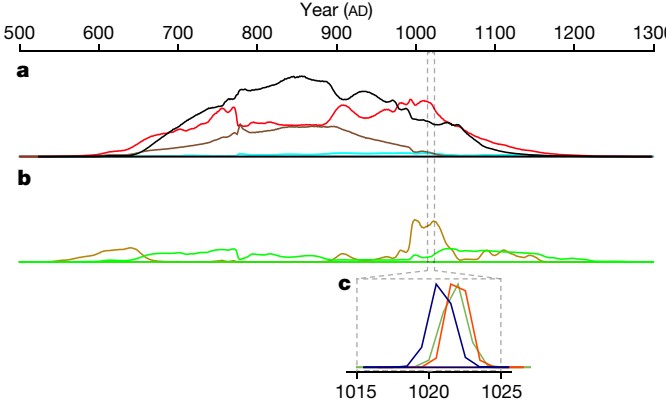

**Fig. 1 | Date ranges obtained from our wiggle matches in comparison with legacy $^{14}$C data. a**, **b**, Averaged probability density functions for different sample types (Extended Data Fig. 3, Supplementary Note 5 and Supplementary Data 1). **a**, Samples susceptible to inbuilt age. Light blue, whale bone (*n* = 1, uncorrected for marine reservoir effect); red, wood (*n* = 17); brown, burnt wood (*n* = 7); black, charcoal (*n* = 22). **b**, Short-lived samples. Light green, turf or sod from the Norse buildings (*n* = 4); olive, outermost rings and twigs from Norse-modified wood (*n* = 4). **c**, Wiggle-matched probability density functions for the last growth ring of each wood item. Dark green, 4A 59 E3-1; navy, 4A 68 J4-6; orange, 4A 68 E2-2.

cells in the waney edge, one can even determine the precise felling season.

## Precise dating of Norse activity in the Americas

Here we present 127 $^{14}$C measurements, of which 115 were performed at the Centre for Isotope Research (CIO; Groningen), and 12 at the Curt-Engelhorn-Center Archaeometry (CEZA; Mannheim). The samples consisted of 83 individual tree rings from a total of 4 wooden items with find numbers 4A 59 E3-1, 4A 68 E2-2, 4A 68 J4-6 and 4A 70 B5-14 (Extended Data Fig. 5, Supplementary Note 4 and Supplementary Data 2). Unfortunately, the last item is excluded from the remainder of our analysis because it spans only nine years and does not include the AD 993 anomaly and therefore cannot be precisely dated (Supplementary Data 2). Anatomical characteristics such as different numbers of growth rings, varying growth-ring widths and the presence–absence of features such as missing rings show that wood items 4A 59 E3-1, 4A 68 E2-2 and 4A 68 J4-6 come from different trees. Furthermore, they comprise at least two different species, specifically fir, possibly balsam fir (*Abies* cf. *balsamea*), and juniper/thuja (*Juniperus*/*Thuja* type; Extended Data Fig. 6). In addition, the waney edge could be identified in all cases.

The items were found at the locations shown on the site map in Extended Data Fig. 2. The association of these pieces with the Norse is based on detailed research previously conducted by Parks Canada. The determining factors were their location within the Norse deposit and the fact that they had all been modified by metal tools, evident from their characteristically clean, low angle-in cuts[30]. Such implements were not manufactured by the Indigenous inhabitants of the area at the time[30] (Supplementary Note 4).

Our individual $^{14}$C results are consistently better than ±2.5‰ (1$\sigma$), with some averaged results better than ±1.5‰ (about 12 $^{14}$C yr). Our corpus of replicated measurements is consistent with statistical expectation, and no statistically significant offset (5.1 ± 7.9 $^{14}$C yr, 1$\sigma$) was evident between the two $^{14}$C facilities involved (Supplementary Data 2).

Two steps are used to determine the exact cutting year of each piece of wood. First, the range of possible dates for the waney edges is obtained by standard $^{14}$C wiggle matching against the Northern

Hemisphere calibration curve, IntCal20 (ref. [23]). Here we use the D_Sequence function in the software OxCal (ref. [31]) to match the full $^{14}$C time-series for each item. The resultant 95% probability (2$\sigma$) ranges for the waney edges all lie between AD 1019 and AD 1024 (Fig. 1c). This indicates that the AD 993 anomaly should be present in each of the wood pieces 26 to 31 years before they were cut. In our numbering system, this corresponds to rings −31 to −26, where the waney edge is assigned to be 0, the penultimate ring is assigned to be −1, and so forth.

A second step is then used to determine the exact cutting year of each item. This process hinges on identifying the precise ring in which the AD 993 anomaly is found, and hence the precise date of the waney edge. For this purpose, we use the Classical $\chi^2$ approach[20,32] to match the $^{14}$C data from the six rings (−31 to −26) most likely to contain the AD 993 anomaly against a second Northern Hemisphere reference (henceforth B2018)[28]. This dataset is preferred because the AD 993 anomaly is less distinct in the smoothed IntCal20 curve (Fig. 2). The six-ring subsets are compared with B2018 such that $\chi^2$ becomes minimal for the cutting date of each item. The matches are conducted over a range for each waney edge of AD 1016−1026 (Fig. 2a).

The optimal $\chi^2$ value for goodness-of-fit for the waney edge in all three cases is AD 1021 (Fig. 2a). While other solutions pass the $\chi^2$ test at 95% probability (AD 1022 for 4A 59 E3-1; AD 1022 for 4A 68 E2-2; AD 1019, AD 1020 and AD 1022 for 4A 68 J4-6), the ideal positioning for the precipitous drop in $^{14}$C years in each case is when ring −29 corresponds to AD 992 (inset of Fig. 2b). Furthermore, the formation of a small band of earlywood cells in 4A 68 J4-6 indicates a felling season in spring (Extended Data Fig. 7a). The felling season of 4A 68 E2-2 is summer/autumn (Extended Data Fig. 7b). Past polyethylene glycol (Methods) consolidation hinders determination of the felling season of 4A 59 E3-1.

Our result of AD 1021 for the cutting year constitutes the only secure calendar date for the presence of Europeans across the Atlantic before the voyages of Columbus. Moreover, the fact that our results, on three different trees, converge on the same year is notable and unexpected. This coincidence strongly suggests Norse activity at L'Anse aux Meadows in AD 1021. Further evidence reinforces this conclusion. First, the modifications are extremely unlikely to have taken place before this year, because the globally observed sudden decrease in $^{14}$C values is evident in ring −29. Second, the probability that the items would have been modified at a later stage is also negligible. This is largely because of the fact that they all had their waney edges preserved. This layer would almost certainly have been stripped off during water transport, so the possibility of driftwood is effectively discounted[33]. Further, the Norse would have had no need to reclaim deadwood because fresh wood was abundant in the region at the time[13]. Finally, if it were scavenged material, the probability that all three items would exhibit precisely the same amount of inbuilt age would be vanishingly small.

The Icelandic sagas suggest that the Norse engaged in cultural exchanges with the Indigenous groups of North America[34]. If these encounters indeed occurred, they may have had inadvertent outcomes, such as pathogen transmission[7], the introduction of foreign flora and fauna species, or even the exchange of human genetic information. Recent data from the Norse Greenlandic population, however, show no evidence of the last of these[8]. It is a matter for future research how the year AD 1021 relates to overall transatlantic activity by the Norse. Nonetheless, our findings provide a chronological anchor for further investigations into the consequences of their westernmost expansion.

## Conclusions

We provide evidence that the Norse were active on the North American continent in the year AD 1021. This date offers a secure juncture for late Viking chronology. More importantly, it acts as a new point-of-reference

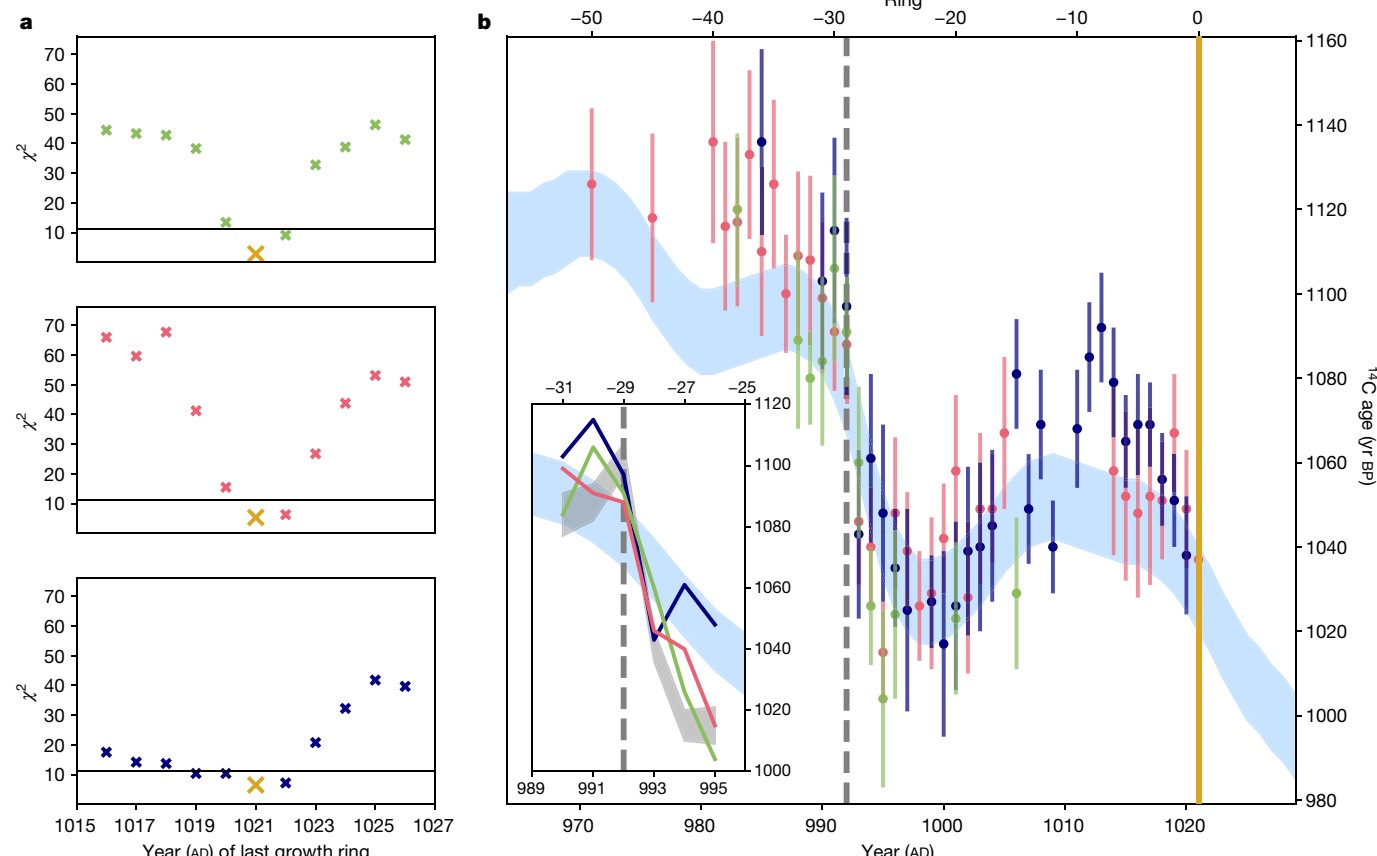

**Fig. 2 | Exact date matches obtained from the $\chi^2$ tests.** The wood items are identified as follows: 4A 59 E3-1 (dark green); 4A 68 J4-6 (navy); 4A 68 E2-2 (orange). **a**, Outputs of the $\chi^2$ test against B2018 (ref. [28]; d.f. = 5, critical value = 11.07, 95% probability), where the gold cross marks the year of best fit for the waney edge. **b**, All of the $^{14}$C data from 4A 59 E3-1 ($n = 12$, $1\sigma$), 4A 68 J4-6 ($n = 35$, $1\sigma$) and 4A 68 E2-2 ($n = 29$, $1\sigma$) superimposed on IntCal20 (light blue, $1\sigma$). Inset: detail of the $^{14}$C results (error bars omitted for legibility) for growth rings −31 to −26 against B2018 (grey, $1\sigma$)[28] and IntCal20 (light blue).

for European cognisance of the Americas, and the earliest known year by which human migration had encircled the planet. In addition, our research demonstrates the potential of the AD 993 anomaly in atmospheric $^{14}$C concentrations for pinpointing the ages of past migrations and cultural interactions. Together with other cosmic-ray events, this distinctive feature will allow for the exact dating of many other archaeological and environmental contexts.

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

# Article

## Methods

### Sampling

After careful examination of the transversal and radial sections of the wood, and ring counting, individual samples were collected under a microscope for annual-ring measurement using a steel blade, following the standard procedure for cleaving tree rings. Sample extraction started at the waney edge. For each wood item, the sample of the waney edge was given the number 0, the second-to-last ring was given the number −1, and so forth.

### Sample preparation and measurement

The tree-ring samples were cut into small fragments again using a steel blade. All of the wood samples were chemically pretreated and analysed at CIO, Groningen. For independent control, 12 of the samples were also chemically pretreated and analysed at CEZA, Mannheim. CEZA and CIO recently took part in a multi-laboratory intercomparison exercise to ensure the effectiveness of their pretreatment protocols in which tree-ring samples of unknown age were pretreated to α-cellulose and then analysed for [14]C concentration by AMS[35].

### Procedures at CIO, University of Groningen

The first step involves pretreating the samples to α-cellulose, the most rigid and immobile fraction of the wood[36]. The method has previously been described in full[37]. In brief, the wet chemistry involves a series of strong solutions of acid–base–acid and an oxidant, with rinses to neutrality using deionized and ultrapure water after each step. The samples are then either freeze-dried or air-dried at room temperature for 72 h. To eliminate the additive polyethylene glycol (PEG), which was present in all wood items except 4A 68 E2-2, the aqueous pretreatment is preceded by placement of the samples in ultrapure water at 80 °C for 36 h. This latter step builds on past studies of this contaminant[38–40]. In cases where the starting weight was <20 mg, and the wood was not treated with PEG, the holocellulose protocol used at CIO was deemed sufficient[37].

Aliquots (about 5 mg, where possible) of the (alpha-)cellulosic product are weighed into tin capsules for combustion in an elemental analyser (IsotopeCube, Elementar). A small amount of the $CO_2(g)$ released is directed into an isotope ratio mass spectrometer (Isoprime 100) for determination of the stable isotope ratios of C and N, but the majority is cryogenically trapped into Pyrex rigs and reduced to graphite under a stoichiometric excess of $H_2(g)$ over an $Fe(s)$ catalyst. The graphite (about 2 mg) is subsequently pressed into $Al(s)$ cathodes for measurement by AMS (MICADAS, Ionplus). The data were refined using BATS 4.0 and stored in FileMaker Pro 14.6.0. For quality control purposes, full pretreatment and radioisotope measurements were concurrently conducted on known-age standards (for example, tree-ring material from AD 1503, UK) and background wood (Pleistocene deposit Kitzbühel, Austria). Community-wide isotope ratio mass spectrometry and AMS standards (for example, National Institute of Standards and Technology oxalic acid II, Merck caffeine, and International Atomic Energy Agency C7 and C8) were used to validate the isotope measurements.

### Procedures at CEZA, Mannheim

Samples MAMS-45877–45879 and MAMS-47884–47886 are pretreated as holocellulose and are pretreated using the acid–base–acid method (acid/base/acid, HCl/NaOH/HCl) followed by bleaching with $NaClO_2$ to extract the cellulose[41]. The second batch of samples (MAMS-50444–50449) is pretreated as alpha-cellulose following the protocol used by CIO described above. PEG contamination is removed in the same way as at CIO by washing in hot ultrapure water. The cellulose is combusted to $CO_2$ in an elemental analyser. $CO_2$ is then converted catalytically to graphite. [14]C is analysed in-house using an AMS instrument of the MICADAS type. The isotopic ratios ([14]C/[12]C of samples, calibration standard oxalic acid II), blanks and control standards are measured simultaneously in the AMS. [14]C ages are normalized to $\delta^{13}C = −25‰$ (ref. [42]), where $\delta^{13}C = (((^{13}C/^{12}C)_{sample}/(^{13}C/^{12}C)^{standard}) − 1) \times 1{,}000$.

### Models in the program OxCal

All models employ OxCal 4.4 and use its standard Metropolis–Hastings Markov chain Monte Carlo algorithm and default priors[31]. The code for these models is provided in Supplementary Note 5 and in the repository https://github.com/mwdee/LAM1021.

### Averaging

Averages are produced for each sample type using the Sum function in OxCal 4.4. In each case, all of the relevant [14]C dates are included in bounded phases. The main prior information used by this model is that each date is assumed to be part of a defined group[31].

### Wiggle matching

[14]C data for each beam are wiggle matched against the IntCal20 calibration curve in OxCal 4.4 using its D_Sequence function[31]. All models show high convergence and run to completion.

### Pattern matching using the $\chi^2$ test

The measured [14]C concentrations of tree-ring samples are matched to a reference curve through the classical statistical method of the $\chi^2$ test[20,22], using the following $\chi^2$ function:

$$\chi^2_{(x)} = \sum_{i=1}^{n} \frac{(R_i - C(x - r_i))^2}{\delta R_i^2 + \delta C(x - r_i)^2}$$

Here $R_i \pm \delta R_i$ are the measured [14]C dates of the sample; $C(x − r_i) \pm \delta C(x − r_i)$ are the [14]C concentrations of the reference curve for the year $(x − r_i)$, where $r_i$ are the tree-ring numbers of the samples analysed; and $x$ is a trial age for the waney edge. Measured dates are matched to the reference data (that is, either higher or lower) in such a way that the $\chi^2$ becomes minimal for a certain value of $x$, which is the best estimate for the felling date of the tree[20]. To match the event accurately, a reference dataset is needed that has single-year resolution. We use B2018 as this reference, which combines many annual [14]C results for the years relevant to this study[28]. The pattern-matching analyses are predominantly carried out using Python 3 in Jupyter Notebook 6.3.0. The results on each of the wood items studied are shown in Fig. 2.

### Wood taxonomy

From the three main fragments of wood (4A 59 E3-1, 4A 68 E2-2 and 4A 68 J4-6), thin sections are prepared under a stereomicroscope with magnifications of up to 50×. They are cut in three directions (transverse, radial and tangential). As the wood was dry, the sections had to be soaked in soapy water to get rid of air bubbles and to be able to see the diagnostic anatomical features. The slides are examined under a transmitted light microscope with magnifications up to ×400 and identified with the help of relevant literature[43–45]. The three samples do not have any vessels, and therefore must be softwood from conifer species. The most important characteristics for identification are the lack of resin canals, the height of the rays (on average much lower in 4A 68 J4-6 than in the other two samples) and the type, number and distribution of the crossfield pits. Also, presence/absence of axial parenchyma, the shape of the ray cells in crossfields, the pitting in side walls and end walls of the ray cells, and the geographical provenance are taken into account. As wood sample 4A 68 J4-6 is compression wood (reaction wood on the lower side of branches and leaning stems), the distinction between cupressoid and taxoidoid pits cannot be made. The identification for this sample is therefore uncertain with juniper and thuja as possible candidates (*Juniperus/Thuja* type). The other two samples are identified with confidence as fir (*Abies*). Within this genus further identification is impossible, but balsam fir (*A. balsamea*), a very common North American species, would be a good match.

## Reporting summary

Further information on research design is available in the Nature Research Reporting Summary linked to this paper.

## Data availability

All of the data that support the findings of this study are available in the main text or Supplementary Information. Source data are provided with this paper.

## Code availability

The codes of the OxCal models are provided in the Supplementary Information and in the repository https://github.com/mwdee/LAM1021.

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

**Acknowledgements** This work was funded by the European Research Council (grant 714679, ECHOES). M.K., A.S., P.E. and M.W.D. were supported by this grant. We thank Parks Canada for providing samples; the CIO staff, especially S. W. L. Palstra, D. van Zonneveld, R. Linker, S. de Bruin, R. A. Schellekens, P. Wietzes-Land, D. Paul, H. A. J. Meijer, J. J. Spriensma, H. G. Jansen, A. Th. Aerts-Bijma and A. C. Neocleous; and R. Doeve, E. van Hees, A. J. Huizinga, B. J. S. Pope and J. Higdon for their help and support.

**Author contributions** M.W.D. conceived the idea, directed the research and co-wrote the paper; M.K. helped to design the research, conducted most of it and co-wrote the paper; B.L.W. was principal advisor on archaeology and sagas; C.L. advised on archaeology; A.S. mainly performed the $\chi^2$ analyses; P.D. advised on tree-ring anatomy; K.J. took samples; S.L. conducted pretreatments (Mannheim); P.E. conducted pretreatments (Groningen); P.M.L. and V.F. advised on archaeology and palaeoecology; C.V. analysed wood taxonomy; R.F. oversaw AMS analyses (Mannheim). All co-authors contributed to the final draft of the manuscript.

**Competing interests** The authors declare no competing interests.

**Additional information**
**Correspondence and requests for materials** should be addressed to Margot Kuitems or Michael W. Dee.

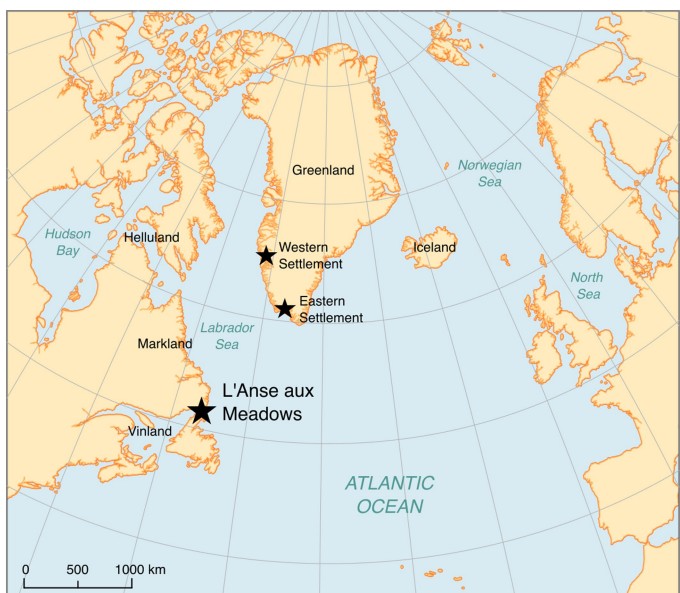

**Extended Data Fig. 1 | North Atlantic regions explored by the Norse.** LAM lies on the Northern Peninsula of Newfoundland. The map shows the main settlements on Greenland from where the Norse embarked, and the regions they named Helluland, Markland and Vinland. Map: R. Klaarenbeek.

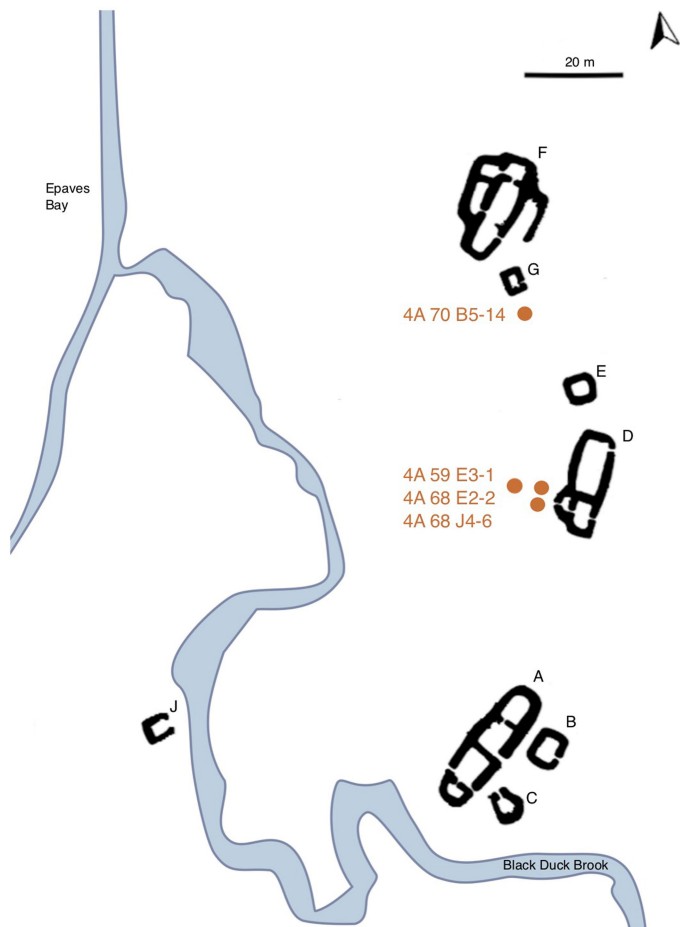

**Extended Data Fig. 2 | Schematic overview of the site (after Wallace 2003)[2] and origin of our samples.** Indicated are the contours of different Norse structures (A–J) and the locations (brown) at which the wood items were found that are used in the current study.

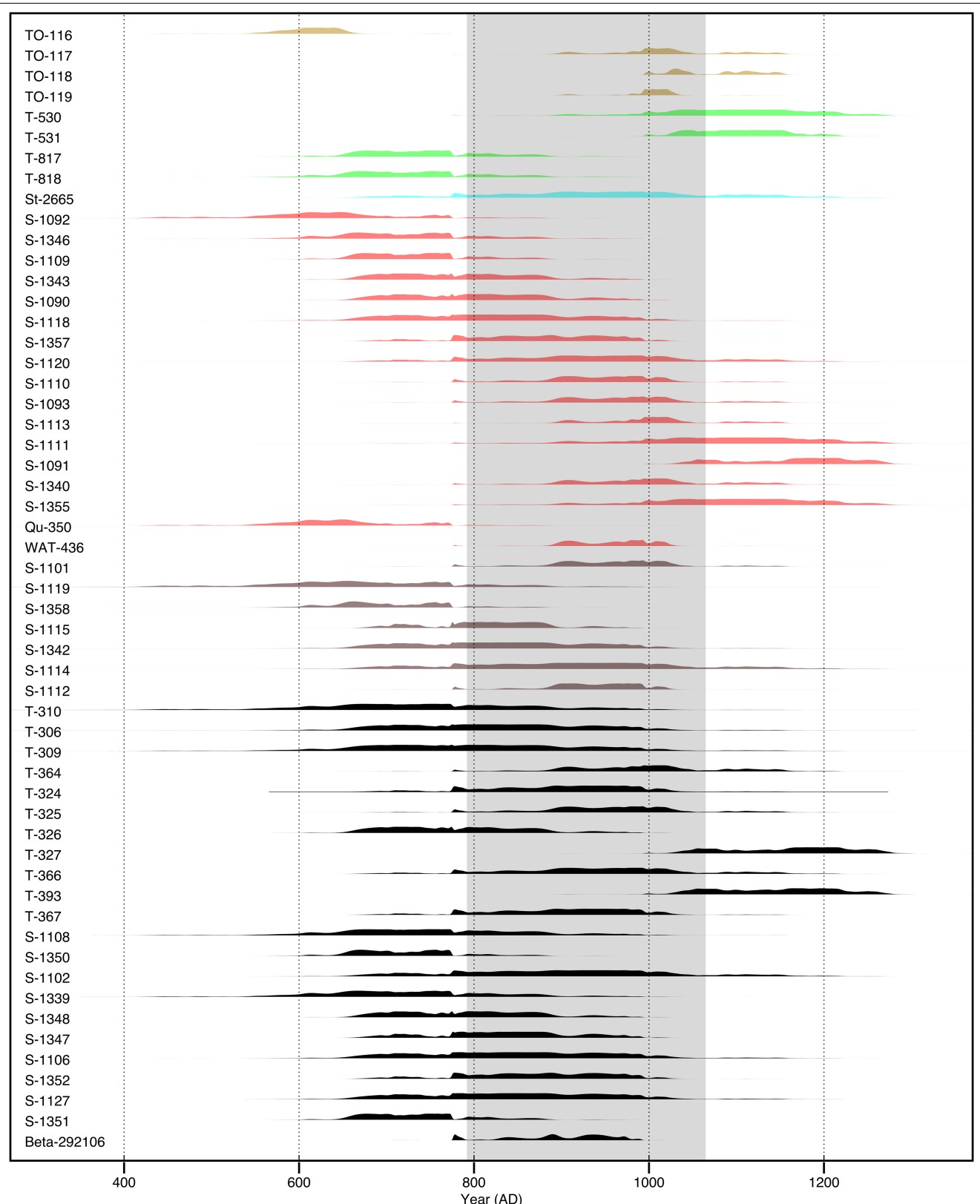

**Extended Data Fig. 3 | The 55 legacy ¹⁴C dates on Norse contexts at LAM.**
Samples susceptible to inbuilt age: light blue, whale bone (*n* = 1, uncorrected for Marine Reservoir Effect); red, wood (*n* = 17); brown, burnt wood (*n* = 7); black, charcoal (*n* = 22). Short-lived samples: light green, turf or sod samples from the walls of the Norse buildings (*n* = 4); olive, outermost rings and twigs from Norse-modified wood (*n* = 4). See Supplementary Data 1.

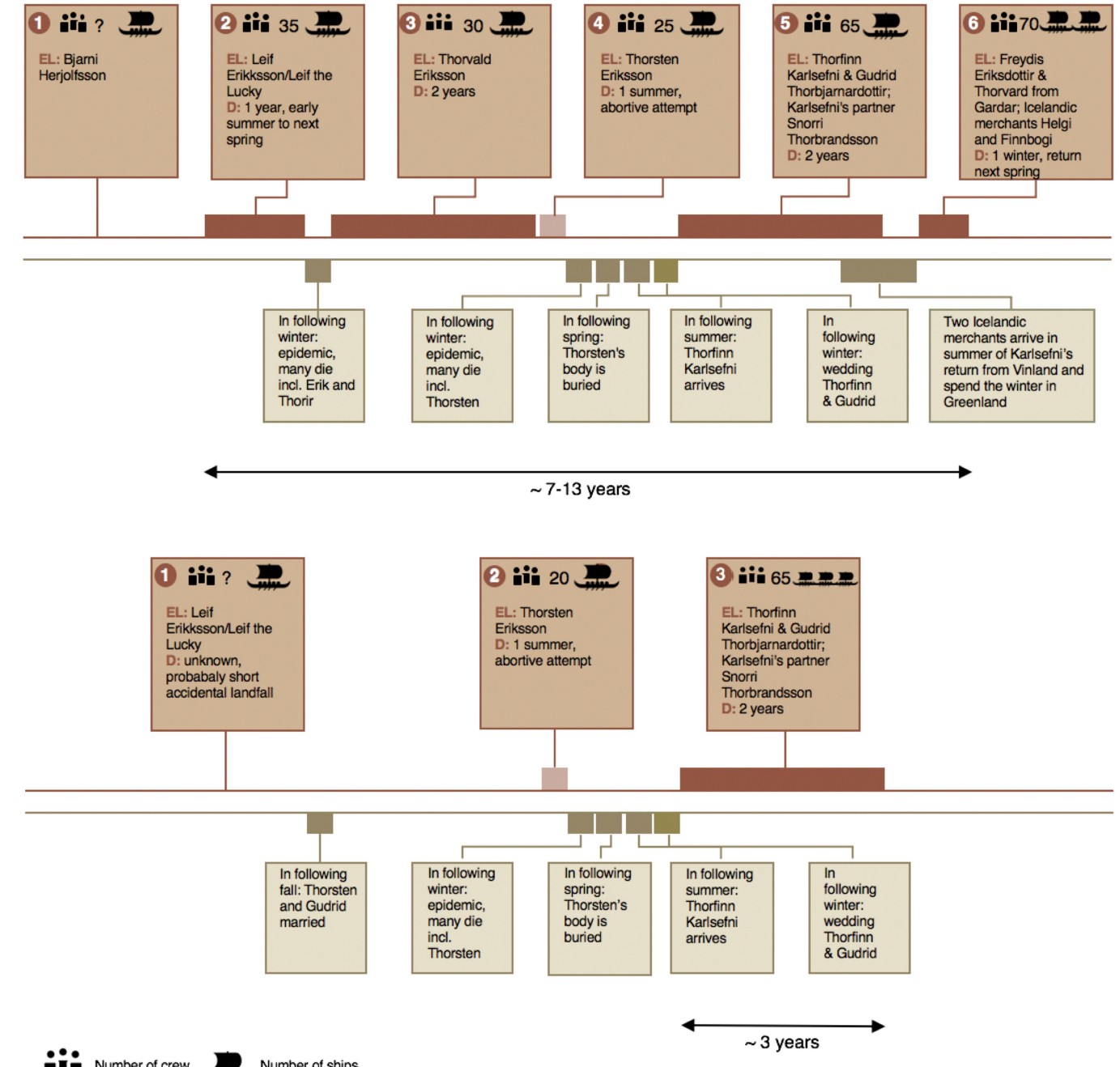

**Extended Data Fig. 4 | Overview of the number and order of the different voyages by the Norse to the Americas based on the information from the Sagas.** Indicated for each voyage are the expedition leader (EL), the duration (D), the number of attending crew and the number of ships. Top, summary of the information from the *Saga of the Greenlanders*, which indicates that the number of winters spent at Vinland is seven. Given the short sailing seasons and the impossibility of making round trips between Greenland and Vinland in one year, the time between the first arrival of the Norse at Vinland and their ultimate return is estimated to be about thirteen years; bottom, summary of the information from the *Saga of Erik the Red*, with the estimated minimum time between the first arrival of the Norse at Vinland and their ultimate return, amounting to about three years.

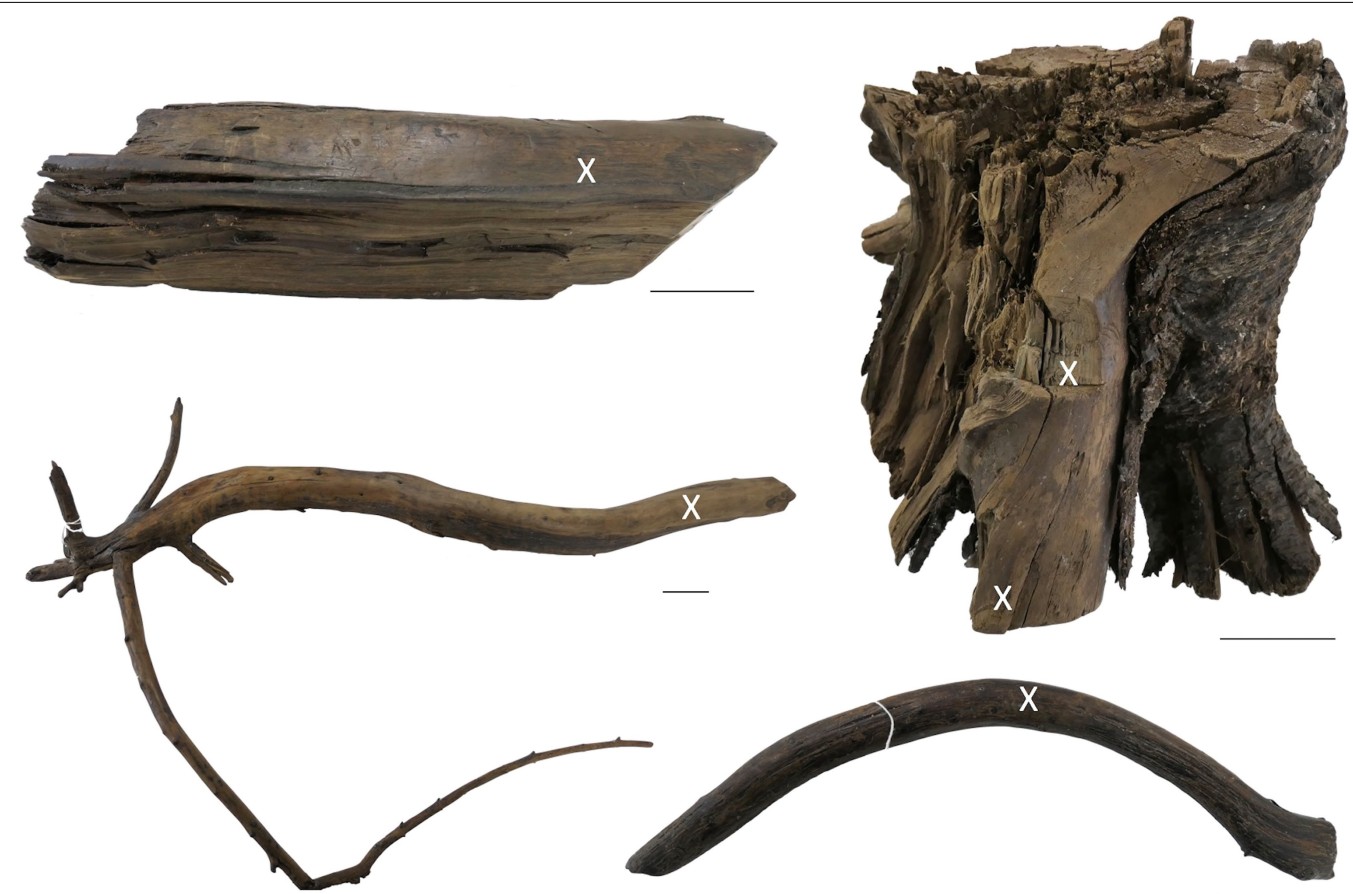

**Extended Data Fig. 5 | Pictures of the wood items studied.** White X indicates the location from where samples were taken. The black bars represent 5 cm. Top left, 4A 59 E3-1; top right, 4A 68 E2-2; bottom left, 4A 68 J4-6; bottom right, 4A 70 B5-14. Photos: M. Kuitems.

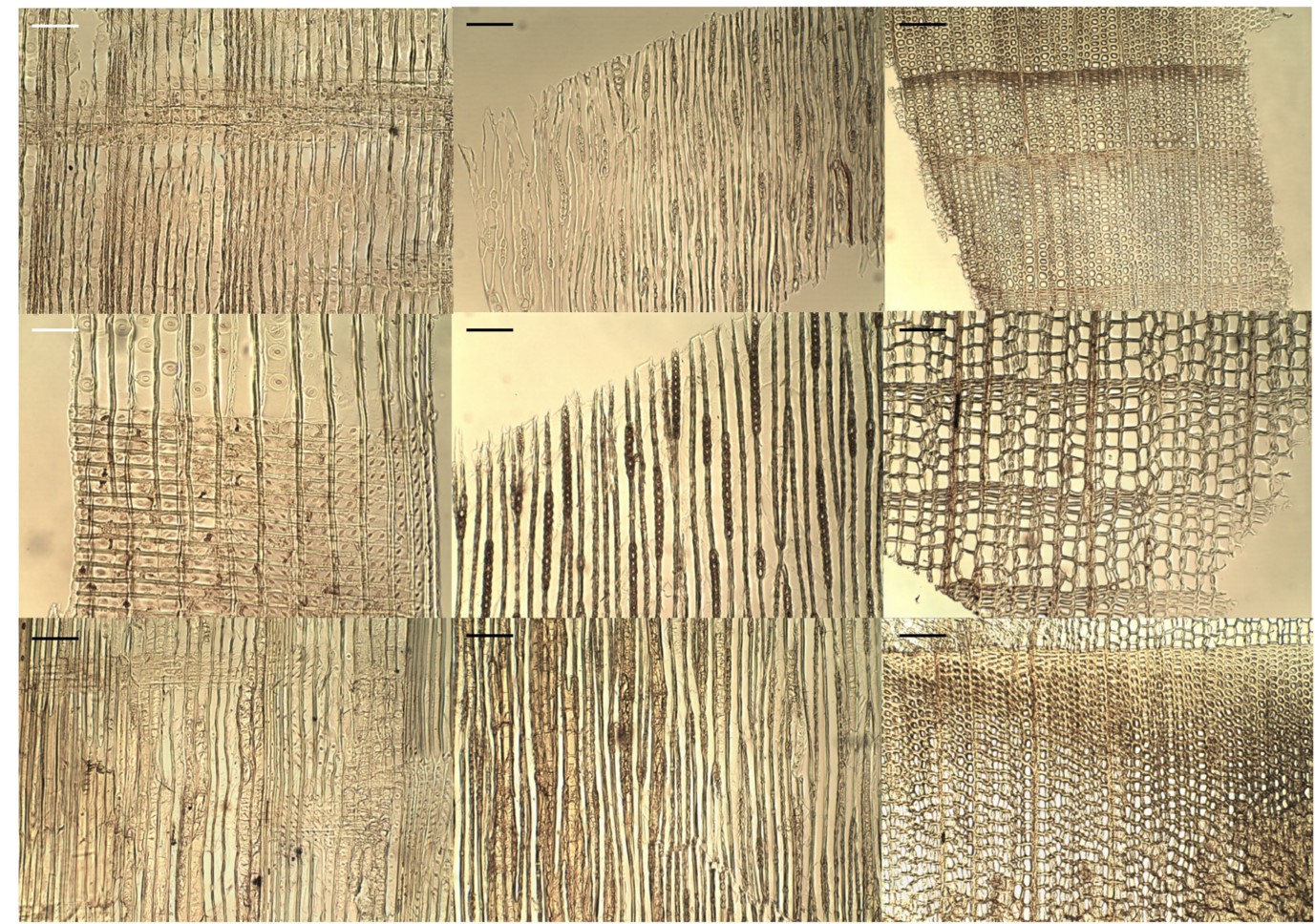

**Extended Data Fig. 6 | Microscope pictures of the thin slices from the wood samples studied.** The white bars represent 0.05 mm, the black bars 0.1 mm. From left to right: radial, tangential and transversal sections of respectively: top, 4A 68 J4-6; middle, 4A 68 E2-2; bottom, 4A 59 E3-1. Photos: M. van Waijjen, BIAX Consult.

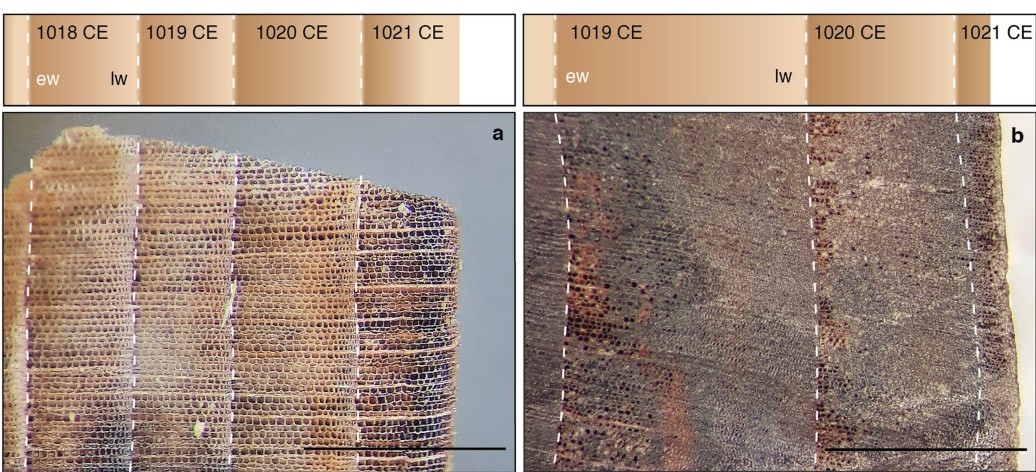

**Extended Data Fig. 7 | Microscopic depiction of the felling season of the waney edge.** The black bars represent 1 mm. ew = early wood, which is formed during the first stage of the growth year; lw = late wood, which is formed at the end of the growth season. **a**, Wood item 4A 68 E2-2; **b**, Wood item 4A 68 J4-6. Photos: P. Doeve.

Margot Kuitems

# Reporting Summary

## Statistics

For all statistical analyses, confirm that the following items are present in the figure legend, table legend, main text, or Methods section.

| n/a | Confirmed | |
|---|---|---|
| ☐ | ☒ | The exact sample size (*n*) for each experimental group/condition, given as a discrete number and unit of measurement |
| ☐ | ☒ | A statement on whether measurements were taken from distinct samples or whether the same sample was measured repeatedly |
| ☐ | ☒ | The statistical test(s) used AND whether they are one- or two-sided *Only common tests should be described solely by name; describe more complex techniques in the Methods section.* |
| ☒ | ☐ | A description of all covariates tested |
| ☐ | ☒ | A description of any assumptions or corrections, such as tests of normality and adjustment for multiple comparisons |
| ☐ | ☒ | A full description of the statistical parameters including central tendency (e.g. means) or other basic estimates (e.g. regression coefficient) AND variation (e.g. standard deviation) or associated estimates of uncertainty (e.g. confidence intervals) |
| ☒ | ☐ | For null hypothesis testing, the test statistic (e.g. *F*, *t*, *r*) with confidence intervals, effect sizes, degrees of freedom and *P* value noted *Give P values as exact values whenever suitable.* |
| ☐ | ☒ | For Bayesian analysis, information on the choice of priors and Markov chain Monte Carlo settings |
| ☒ | ☐ | For hierarchical and complex designs, identification of the appropriate level for tests and full reporting of outcomes |
| ☒ | ☐ | Estimates of effect sizes (e.g. Cohen's *d*, Pearson's *r*), indicating how they were calculated |

*Our web collection on statistics for biologists contains articles on many of the points above.*

## Software and code

Policy information about availability of computer code

| | |
|---|---|
| Data collection | Data were collected by authors from Canada (Government of Canada, Parks Canada Agency), Netherlands (University of Groningen, BIAX Consult, Laboratory for Dendrochronology at BAAC and Rijksdienst voor het Cultureel Erfgoed) and Germany (CEZA Mannheim) using the software below. Radiocarbon dates were acquired on a MICADAS AMS system using BATS software 4.0. Raw data is stored in FileMaker Pro 14.6.0 |
| Data analysis | OxCal 4.4 using standard Metropolis-Hastings Markov Chain Monte Carlo (MCMC) algorithm and default priors Canadian Archaeological Radiocarbon Database 2.0 Python 3 in Jupyter Notebook 6.3.0 Inkscape 1.0.1 Further details on the methods and relevant codes are included in the manuscript and the Supplementary Information and are available on this repository: https://github.com/mwdee/LAM1021 |

For manuscripts utilizing custom algorithms or software that are central to the research but not yet described in published literature, software must be made available to editors and reviewers. We strongly encourage code deposition in a community repository (e.g. GitHub). See the Nature Portfolio guidelines for submitting code & software for further information.

## Data

Policy information about availability of data

All manuscripts must include a data availability statement. This statement should provide the following information, where applicable:
- Accession codes, unique identifiers, or web links for publicly available datasets
- A description of any restrictions on data availability
- For clinical datasets or third party data, please ensure that the statement adheres to our policy

All the data that support the findings of this study are available in the main text, Extended Data files or Supplementary Information files.

# Field-specific reporting

Please select the one below that is the best fit for your research. If you are not sure, read the appropriate sections before making your selection.

☒ Life sciences ☐ Behavioural & social sciences ☐ Ecological, evolutionary & environmental sciences

For a reference copy of the document with all sections, see nature.com/documents/nr-reporting-summary-flat.pdf

# Life sciences study design

All studies must disclose on these points even when the disclosure is negative.

| | |
|---|---|
| Sample size | We did not rely on statistical methods to predetermine sample sizes. Sample sizes in this study on an archaeological site were limited by the availability of excavated wood specimens and numbers of growth rings they contained. |
| Data exclusions | Data was obtained on sample 4A 70 B5-14 (see Source Data Fig. 2). However, no pattern matching was attempted for this sample, because the 993 CE anomaly was clearly not evident and therefore it could not be precisely dated. Two sample measurements that failed the community's standard chi square statistical test for congruity were excluded from analysis (see below). Further details are provided in the manuscript and Extended Data. |
| Replication | 34 samples were pretreated and radiocarbon dated more than once (as duplicates, triplicates and one quadruplicate). 32 successfully passed the community's standard (chi square) statistical test for congruity (94.1 vs 95.4% expected probability). On 12 occasions, the repeated pretreatments and measurements were conducted by another radiocarbon facility (CEZA Mannheim). We concentrated the replicates around the most distinctive features of the radiocarbon record to ensure our key findings were robust. Further details are provided in the manuscript and Source Data Fig. 2. |
| Randomization | The data for this study were acquired in no specific order or grouping, and all individual results were assessed against the same pretreatment, IRMS, and AMS quality control standards. |
| Blinding | The 12 samples that were were sent for replication to CEZA Mannheim were supplied without any information about the origin or expected age. |

# Reporting for specific materials, systems and methods

We require information from authors about some types of materials, experimental systems and methods used in many studies. Here, indicate whether each material, system or method listed is relevant to your study. If you are not sure if a list item applies to your research, read the appropriate section before selecting a response.

## Materials & experimental systems

| n/a | Involved in the study |
|---|---|
| ☒ | Antibodies |
| ☒ | Eukaryotic cell lines |
| ☐ | ☒ Palaeontology and archaeology |
| ☒ | Animals and other organisms |
| ☒ | Human research participants |
| ☒ | Clinical data |
| ☒ | Dual use research of concern |

## Methods

| n/a | Involved in the study |
|---|---|
| ☒ | ChIP-seq |
| ☒ | Flow cytometry |
| ☒ | MRI-based neuroimaging |

## Palaeontology and Archaeology

| | |
|---|---|
| Specimen provenance | The four samples used in this study had Canadian Heritage Parks Canada catalogue numbers 4A 59 E3-1, 4A 68 E2-2, 4A 68 J4-6 and 4A 70 B5-14. A Collections and Transfer Registration form was authorized by Parks Canada Agency official Kevin Jenkins and signed upon reception by researcher Dr Margot Kuitems on 9/11/2018 at Halifax, Nova Scotia. |

| Specimen deposition | The main items were returned to the official Parks Canada Storage after sampling. The excess material after analysis is held at CIO, Groningen. Requests for sample material can be made to the corresponding authors. |
|---|---|
| Dating methods | The radiocarbon dates were obtained on alpha-cellulose extracts taken from the above-mentioned samples at CIO (Groningen) and CEZA (Mannheim). The pretreatment, measurement and analytical methods employed by both laboratories are all described in full in the main text. Data quality was ensured by the use of internal and external standards (e.g. tree ring material from 1503 CE, UK; background wood from a Pleistocene deposit Kitzbühel, Austria; NIST oxalic acid II; Merck™ caffeine; IAEA C7 and C8). Results were calibrated using OxCal 4.4 against IntCal20. |

☒ Tick this box to confirm that the raw and calibrated dates are available in the paper or in Supplementary Information.

| Ethics oversight | Approval to undertake the study was given by the University of Groningen. Approval to take and analyse the samples, and publish the results, was obtained from Government of Canada, Parks Canada Agency. |
|---|---|

Note that full information on the approval of the study protocol must also be provided in the manuscript.

nature portfolio | reporting summary

March 2021

3