## [Peer Review File · Nature]

Manuscript Title: Evidence for European Presence in the Americas in 1021 CE

Reviewer Comments & Author Rebuttals

Reviewer Reports on the Initial Version:

Referee #1 (Remarks to the Author):

Dear authors,

I congratulate the authors for the very nice work they present in their manuscript! I strongly recommend a publication after some rather minor revisions. The authors propose the first accurate and precise date for early European presence (by Vikings) in North America. This is a most significant finding which verifies and supports for the first time precisely the dates given by the famous 13–15th century Icelandic Sagas. The work proves an early settlement in L'Anse aux Meadows in 1021 AD, far earlier than the discovery of America by Columbus, what is certainly of highest interest to a broad audience.

The researchers applied state-of-the-art methods using high precision radiocarbon dating. While the method using radiocarbon events for precise and accurate dating is not applied for the first time, it is certainly the most significant date produced so far in Archeology using this new method. The results are highly significantly and securely verified (several objects with replications by 2 laboratories).

The statistical approach applied are in correct, but is not always conclusive and clear (see also comments). Uncertainties are not given (while they must be very small and as a consequence give a very robust and clear result).

The introduction is ok, but I am missing a few details that I found myself (not an Archeologist) relatively quickly in the internet. Previous assumptions should clearly be mentioned (though primarily relying on a Saga). A discussion of the implications of the presented dating (on previous assumptions) is missing and I think that this needs to be added.

Best
Lukas Wacker

Comments:

You may consider using an updated title mentioning Vikings or Norse

Line 37: At least from what we know - a reference would be good to have here.

Line 50: I think this is a matter of interpretation.

Line 59: What do you mean by limitation of this technique? What is "this technique"?

Line 72: Less than 1-2 permil (fluctuations in Intcal are due to measurement noise primarily

Line 79: ref 20 should also be given here at the end of the sentence.

Line 79: Why Moreover? Isn't this sentence basically telling the same as the sentence before?

Results and discussion: you start the section with a sampling and method part...

Lines 93 - 98: This sounds like an unmotivated enumeration.

Line 106: What is then the offset? Give a scientific measure!

Line 122: This sentence is not clear... you calculate χ^2 for different positions in time, right? How many years were taken for the tests (degree of freedom is important).

Line 125: What is ring number 29? Counting backwards from the waxy edge? Please explain!

Line 126: You haven't been very lucky that in each case ring 29 gives the best match! It looks like that in each case several positions are statistically possible. Please give us some numbers on the precision. Also, you probably could calculate a test, where you assume a contemporary waxy edge (and in fig 2c it looks actually different - why?)

Figure 3a seems redundant... is included in figure 3c that is well readable. Also, why do you only

plot the blue curve in 995 and not the others?

Figure 3b: What is the horizontal line? Is this the 95% confidence level? What are the degrees of freedom?

Line 140: You say that the only secure result is the year 1021 - I think this is not correct! For each of the 3 samples it looks like you get something like ± 2 years. Maybe combining the 3 under the assumption of a con-temporal wane edge you may get something like ± 1 year.

Line 142: you say the result is Striking and unexpected; but why? Because statistically it is unexpected even assuming a con-temporal wane edge?

Line 143: are you sure with the precisely 1021? Maybe you should add a short discussion whether the samples could also have been transported to the location or not. Are the samples potentially from a different location and/or from previous years?

Line 145: You say that the ring 28 is unambiguously the year 993. That is certainly wrong. Only in the hole context (together with ring 29) you have the indication that it is the year 993.

Line 307ff: you never say anything about the numbers of rings compared to the IntCal curve. Did you take all the samples measured for the match or just a selection?

Extended Data Table 2: Ring numbers are given as negative values - what is probably a good idea as you count from the outside (in contrast to dendrochronologists). In the text, however, you give positive number. That is confusing and should be changed.

Some of the extended data figures (eg Fig 2, 5 and 6) can potentially go into supplementary information.

Referee #2 (Remarks to the Author):

This paper combines dendrochronology, ^{14}C wiggle matching and chi-square approaches in order to establish that the Norse settlement of L'Anse aux Meadows in Newfoundland was definitively occupied in 1021 CE. In particular, it relies on identification of the 993 CE spike in atmospheric ^{14}C within the tree-ring sequences of three wooden objects that exhibit cut marks from metal tools. Given that the Indigenous peoples of Newfoundland and Labrador used stone tools at this time, these objects are considered unambiguous evidence of Norse occupation. Straightforward tree-ring counting from the 993 CE rings to the wane edges of these timbers provided a date of cutting and thus occupation, with all three objects indicating 1021 CE.

It has long been known that the Norse occupation of L'Anse aux Meadows occurred sometime between the late 10th and 12th centuries. A date close to 1000 CE was preferred, but this was based in large part on Icelandic literary sources that are not contemporary with the events described and offer contradictory accounts of specific events. Moreover, earlier radiocarbon dates lacked the precision to improve on a very general assessment of the 10th to 12th century. The provision of a precise and absolute date for occupation at L'Anse aux Meadows is thus both original and important. It anchors the earliest potential contact between European and First Nations peoples in time.

Radiocarbon dating and dendrochronology experts should evaluate specialist aspects of the text. From the perspective of an archaeologist and medievalist, the paper is clearly publishable. Nevertheless, it would benefit from revisions to improve clarity and limit scepticism.

The starting point of the paper's argument is the observation that the three dated timbers were cut with metal tools. However, the only illustrations of the objects (photographs in supplementary extended data material) do not clearly show the cut surfaces. Moreover, there is no detailed explanation of how timber cut with stone and iron tools can be reliably distinguished (however obvious this might be), and no documentation of how the specimens in question fit these criteria. This information is essential, and photographs of at least some of the cut marks should be in the main article, not only the supplementary material.

It is not clear from Figure 3a how the new data match the 993 CE anomaly. The authors should consider an alternative way of demonstrating this match visually - it is not self-evident from the curve plot provided.

In Figure 3 it is also unclear how all three wood objects have the same ring numbering. This is explained in the methods section at the end of the paper (counting back from the waney edge), but it would be helpful were it made equally clear in the main text for the benefit of an interdisciplinary readership. Many will not be familiar with the conventions of dendrochronology.

Given how the methodology is explained in the text (e.g. ascertaining the 993 CE ring in each sample by wiggle matching, and then counting forward to the felling dates of 1021 CE) the authors may wish to reconsider Figure 3b. It shows the results of the 14C chi-square test for 1021 CE, but does not show 993 CE. There is thus a mismatch between the thought process in the text and what is shown in the main figure. It may be clearer to readers if the authors illustrate the 14C attribution of the 993 CE rings (rather than the 1021 rings), and then also provide a drawing or photographs of the tree ring sequence(s) from the 993 CE ring(s) to the waney edge(s), either for all three timbers or one example. Certainly an image of one of the actual tree ring sequences is needed in the main text. At present they occur only in the extended figures.

Slightly more could be made of the seasonality evidence, if positively evaluated by a method expert. Photographic evidence of this seasonality, based on early and late wood development, would then be valuably moved from the extended figures to the main text. If space is short to add these images, the present simple location map (Figure 1) could be reduced and made part of a multi-pane figure.

Slightly more consideration is merited regarding the efficacy of the method used to remove PEG contamination. The approach (soaking in hot water) is noted, but not how comprehensively it works, or how the authors would know if it did not.

Referee #3 (Remarks to the Author):

By applying recent advances in radiocarbon dating methods to four wooden objects retrieved from a Norse settlement in L'Ance aux Meadows, Newfoundland, the authors have been able to date the felling of three trees to 1021 CE. The objects' origin in Norse activities is testified by traces of metal tools on them; the indigenous population did not have metal tools.

The text is easily read, facts and argument are presented clearly and lucidly. The approach seems valid, and the quality of the archaeological data, which is what this reviewer has expert knowledge in, appears to be high. The resulting datings of the wooden samples seem reliable.

Hitherto, the dating of this settlement has relied on saga evidence, artefacts retrieved during excavations, and standard-type radiocarbon datings. While the latter two allow rather wide date ranges, the sagas indicate that periodic settlement in Vinland stretched over c. 3–13 years around 1000 AD.

As carefully stated by the authors, the new dating concerns Norse presence, not the start or end of the Norse settlement. Bearing in mind that sagas do not necessarily mention all Norse journeys to Newfoundland, the site may have been periodically inhabited over a period of more than 13 years. Therefore, the bulk of activity may still lie a decade or two prior to the new dating. Therefore, the new dating does not significantly alter existing assessments of the Norse presence in Newfoundland, although one might now argue that it likely elapsed after the turn of the millennium.

The Norse journeys to north-eastern America is a spectacular but minor episode in Viking History at large, and this reviewer finds that the authors overstate the significance of their findings somewhat. One might be justified in asking whether it is really the case that "Our new date [...] sheds new light on Viking chronology" (lines 30–31)? In this reviewer's estimate, the dating's repercussions for Viking chronology at large are negligible. The authors are advised to limit the cited passage to 'Viking Chronology on the American Continent'.

Likewise, it seems hard to accept that the new dating "provides a definitive tie-point for future research into the consequences of the western expansion of Viking culture" (lines 31-2). The statement might convey the impression that the authors hold their dating to be significant for research into the consequences of Viking settlement in other western lands: Ireland, the British Isles, Normandy, Shetland, the Faroes, Iceland, and Greenland. If that is indeed what they mean, they need to present an argument for their case. Would, for instance, the two studies referred to on this issue (nos. 7–8) have arrived at different conclusions had their authors had access to the new dating? If this is not what they mean, the authors are advised to limit the scope of the cited passage to 'the expansion of Viking culture to the American Continent'.

In conclusion, this reviewer would advise that these and other (see also lines 164–5) rather far-reaching statements are left out altogether and that the authors instead turn their attention to discussing what their new dating has to say for the understanding of the Norse presence in north-eastern America in general and for the site in particular. For example, is it likely that the wooden items from 1021 were produced during one of the stays mentioned in the sagas, or do they more likely result from a subsequent and possibly unrecorded stay?

Referee #4 (Remarks to the Author):

A. This paper represents a very nice piece of work on the dating of the brief Norse occupation of N America using what is essentially a combination of radiocarbon and dendrochronology. It makes particular use of the rapid events in the radiocarbon record assumed to be due to unusual solar activity.

B. The paper is a novel application of this combination of techniques and gives new insight with a precision that is at least an order of magnitude better than previously attainable.

C. The overall methods and approach look appropriate. The data is clearly of high quality both from the inter-comparisons but also from the internal consistency of the data set.

D. In general the use of statistics looks appropriate. However, there is one area which needs looking at in more detail. The Bayesian approach provides the initial analysis and is then backed up with a classical approach using a chi-squared test to find the best fit. This is generally a valid approach but you would expect the lowest chi-squared values to correspond to the highest Bayesian probabilities. I cannot see why sample 4A 68 J4-6 shown in blue in Fig 2c has its highest probability a couple of years earlier than the other samples and yet the same sample has the lowest chi-squared at the same year in Fig 3. This may just be a result of the way the uncertainties in the underlying curve fall relative to the data.

However, more generally I think it is important to argue more carefully for the single year hypothesis. Using either the Bayesian or the classical approaches a number of different fits would be possible - that which is presented is only the most likely. The fact that it also happens to give the same felling date to all the samples is extra support for this - but we probably cannot rule out a year either side of this. By assuming the samples are all the same age the Bayesian approach could give error limits on a combined felling date. Certainly the most likely would be that presented but we probably could not rule out adjacent years at the 95% confidence level.

What is perhaps more compelling as an argument is that fact that all three samples show the rapid drop in the same year with this conclusion.

E. Despite the points above the overall results look very robust: it is very hard to see how this could be significantly wrong in terms of the main conclusion.

F. I would suggest some minor extra discussion on the statistical front. The other area which it would be good to see some discussion in the main text is the origin of the wood in the artefacts. From the arguments of the chronology, I assume the understanding is that these artefacts are of N American wood - made on site. This is never explicitly said and is an important element in the argument. Otherwise these could be material brought from Europe and potentially a few years before.

G. The references seem appropriate

H. The paper is clearly written and the abstract adequately describes the content.

Author Rebuttals to Initial Comments:

To begin, we would like to thank all four reviewers, from various backgrounds, for their insightful and helpful comments. By resolving these queries, we believe we have made the manuscript much clearer and more effective. Please find detailed below (in green) our responses to all of the reviewers' comments.

Referee #1 (Remarks to the Author):

Dear authors,

I congratulate the authors for the very nice work they present in their manuscript! I strongly recommend a publication after some rather minor revisions. The authors propose the first accurate and precise date for early European presence (by Vikings) in North America. This is a most significant finding which verifies and supports for the first time precisely the dates given by the famous 13–15th century Icelandic Sagas. The work proves an early settlement in L'Anse aux Meadows in 1021 AD, far earlier than the discovery of America by Columbus, what is certainly of highest interest to a broad audience.

The researchers applied state-of-the-art methods using high precision radiocarbon dating. While the method using radiocarbon events for precise and accurate dating is not applied for the first time, it is certainly the most significant date produced so far in Archeology using this new method. The results are highly significantly and securely verified (several objects with replications by 2 laboratories).

The statistical approach applied are in correct, but is not always conclusive and clear (see also comments). Uncertainties are not given (while they must be very small and as a consequence give a very robust and clear result).

Best Lukas Wacker

We thank the reviewer for his opinion on the accuracy and importance of our work.

The introduction is ok, but I am missing a few details that I found myself (not an Archeologist) relatively quickly in the internet. Previous assumptions should clearly be mentioned (though primarily relying on a Saga). A discussion of the implications of the presented dating (on previous assumptions) is missing and I think that this needs to be added.

This observation requires a more nuanced response than might be expected. Our work has implications for both understanding the oral traditions (Sagas) and, more broadly, the peopling of the planet. Our main aim was to present a solid, science-based date for this important juncture. In our opinion, the spectrum of interpretations of the Sagas is just too broad (from fact to folklore) to express how such scholars should accommodate our date of 1021 CE. We do state that most Saga analysts position the voyages around the end of the first millennium. But going further than this with the Sagas is beyond the scope of our article. On the other hand, at both the beginning and the end of the article, we place our result in a broader context. We say that it marks the point at which European rulers knew of (inhabited) land over the Atlantic; that it represents humanity's first encirclement of the globe; and that it has significant implications for peoples and ecologies of the Old and New World since their separation with the formation of the Bering Strait 10,000 years earlier.

You may consider using an updated title mentioning Vikings or Norse

Thank you for this suggestion. We did consider several options but we think the current title is still best. The most significant thing about our study is that it relates to the first Europeans in the Americas rather than a specific group of Europeans. In addition, we tried to avoid the word 'Viking' throughout the manuscript, because this term is actually used for an activity and its participants rather than a specific ethnic or cultural group.

Line 37: At least from what we know - a reference would be good to have here.

We have added Wallace 2003 (Reference 9). We still believe it is an acceptable statement as no evidence to the contrary exists.

Line 50: I think this is a matter of interpretation.

Yes, we have updated the sentence. Even though there is some ¹⁴C evidence that the Norse layers LAM date to the Viking Period, we still think it is valid to say ours is the first meaningful scientific date for the site.

Line 59: What do you mean by limitation of this technique? What is "this technique"?
We mean the ^{14}C dating technique. We have changed this to 'this chronometric technique'.

Line 72: Less than 1-2 permil (fluctuations in Intcal are due to measurement noise primarily)
Agreed, done.

Line 79: ref 20 should also be given here at the end of the sentence.
Agreed, done.

Line 79: Why Moreover? Isn't this sentence basically telling the same as the sentence before?
We are adding additional information to the sentence before, so we believe 'moreover' is the appropriate term.

Results and discussion: you start the section with a sampling and method part...
This is an astute observation. However, it relates to only two sentences that outline the number of items and samples analysed. We think shifting them to the Methods section (i.e., after the References) would make the text less clear to the reader.

Lines 93 - 98: This sounds like an unmotivated enumeration.
If by 'unmotivated' the reviewer means 'unnecessary', we actually think this is a list of important information that helps prove the samples were all from different trees, which all had the bark edge preserved.

Line 106: What is then the offset? Give a scientific measure!
Agreed, done.

Line 122: This sentence is not clear... you calculate χ^2 for different positions in time, right? How many years were taken for the tests (degree of freedom is important).
Yes, we have added a sentence to explain that for each wood item a sequence of six rings was taken for the χ^2 tests ($df = 5$). See comment for Line 126 below.

Line 125: What is ring number 29? Counting backwards from the waney edge? Please explain!
Agreed, we have added a sentence to the main text to explain the counting.

Line 126: You haven been very lucky that in each case ring 29 gives the best match! It looks like that in each case several positions are statistically possible. Please give us some numbers on the precision. Also, you probably could calculate a test, where you assume a contemporary waney edge (and in fig 2c it looks actually different - why?)

After the reviews, we realised that our sample-reference matching was not clear, and the χ^2 matches against IntCal20 were inadequate. We have adjusted the text accordingly. The two steps are as follows. (1) The wiggle-matches in OxCal are applied against the smoothed IntCal20 curve to obtain the 95% ranges for the waney edges. These matches make use of *all* the data we obtained on each wood item. The 95% ranges overlap each other (Fig. 2c), and all lie between 1019-1024 CE. This means the 993 CE anomaly has to be between rings -31 and -26. (2) We take the ^{14}C data on these *six rings*, in each case, and χ^2 match them against a more detailed reference (B2018), so that that the χ^2 value was minimised for the waney edge. The new matches with B2018 show that 1021 CE is most likely to be the cutting year for each piece of wood. Although other solutions pass the χ^2 test at 95% probability, the precipitous drop (~ 70 yrs), which is universally accepted to occur between 992-994 CE, is only possible with this positioning. However, we do now list in the main text each calendar year that is possible. Thus, we think there is no further value in averaging the ranges from the OxCal wiggle matches against IntCal20. This would provide an artificially imprecise account of the evidence we have. Finally, an important outcome of our research is the discovery that the cutting years were all the same. If we had simply assumed this, we would be losing information that is clearly empirically present.

Figure 3a seems redundant... is included in figure 3c that is well readable. Also, why do you only plot the blue curve in 995 and not the others? Well observed. As several reviewers found Figure 3a redundant, we have significantly improved the entire figure.

Figure 3b: What is the horizontal line? Is this the 95% confidence level? What are the degrees of freedom?

Agreed. We have added a sentence to explain the line and the degrees of freedom.

Line 140: You say that the only secure result is the year 1021 - I think this is not correct! For each of the 3 samples it looks like you get something like ± 2 years. Maybe combining the 3 under the assumption of a con-temporal waney edge you may get something like ± 1 year.

In the text we now explicitly list all the possible solutions (at 95% probability), whilst explaining that 1021 CE is the most favourable result in each case. So now, as described in the response to Line 126, we contend that arguing for a contemporaneous waney edge and averaging the results would not be the best treatment of the evidence.

Line 142: you say the result is Striking and unexpected; but why? Because statistically it is unexpected even assuming a con-temporal waney edge?

We were not expecting contemporaneous waney edges. The site may have been occupied for multiple years, so different felling dates were certainly possible. As a result, it was striking and unexpected.

Line 143: are you sure with the precisely 1021? Maybe you should add a short discussion whether the samples could also have been transported to the location or not. Are the samples potentially from a different location and/or from previous years?

We do discuss this in the sentences following this statement.

Line 145: You say that the ring 28 is unambiguously the year 993. That is certainly wrong. Only in the hole context (together with ring 29) you have the indication that it is the year 993.

Agreed, fixed.

Line 307ff: you never say anything about the numbers of rings compared to the IntCal curve. Did you take all the samples measured for the match or just a selection?

Please see the extra lines added to the main text as well as the response to Line 126.

Extended Data Table 2: Ring numbers are given as negative values - what is probably a good idea as you count from the outside (in contrast to dendrochnologists). In the text, however, you give positive number. That is confusing and should be changed.

Again, a very useful observation. Agreed, done.

Some of the extended data figures (eg Fig 2, 5 and 6) can potentially go into supplementary information.

Yes, but this journal does not allow figures in the Supplementary Information section.

Referee #2 (Remarks to the Author):

This paper combines dendrochronology, 14C wiggle matching and chi-square approaches in order to establish that the Norse settlement of L'Anse aux Meadows in Newfoundland was definitively occupied in 1021 CE. In particular, it relies on identification of the 993 CE spike in atmospheric 14C within the tree-ring sequences of three wooden objects that exhibit cut marks from metal tools. Given that the Indigenous peoples of Newfoundland and Labrador used stone tools at this time, these objects are considered unambiguous evidence of Norse occupation. Straightforward tree-ring counting from the 993 CE rings to the waney edges of these timbers provided a date of cutting and thus occupation, with all three objects indicating 1021 CE.

It has long been known that the Norse occupation of L'Anse aux Meadows occurred sometime between the late 10th and 12th centuries. A date close to 1000 CE was preferred, but this was based in large part on Icelandic literary sources that are not contemporary with the events described and offer contradictory accounts of specific events. Moreover, earlier radiocarbon dates lacked the precision to improve on a

very general assessment of the 10th to 12th century. The provision of a precise and absolute date for occupation at L'Anse aux Meadows is thus both original and important. It anchors the earliest potential contact between European and First Nations peoples in time.

We are grateful for the reviewer's positive evaluation of our work.

Radiocarbon dating and dendrochronology experts should evaluate specialist aspects of the text. From the perspective of an archaeologist and medievalist, the paper is clearly publishable. Nevertheless, it would benefit from revisions to improve clarity and limit scepticism. The starting point of the paper's argument is the observation that the three dated timbers were cut with metal tools. However, the only illustrations of the objects (photographs in supplementary extended data material) do not clearly show the cut surfaces. Moreover, there is no detailed explanation of how timber cut with stone and iron tools can be reliably distinguished (however obvious this might be), and no documentation of how the specimens in question fit these criteria. This information is essential, and photographs of at least some of the cut marks should be in the main article, not only the supplementary material.

We understand that this information is fundamental to our research. However, the departure point for our project was that these facts were already established and beyond reproach. Indeed, Parks Canada has reports from the 1970-80s showing that the cutmarks were the result of metal tools (e.g. Gleeson 1979). Wallace (2012) contains a comprehensive description of this work. Nonetheless, we have added some sentences to the manuscript to provide the reader with more information on this matter.

It is not clear from Figure 3a how the new data match the 993 CE anomaly. The authors should consider an alternative way of demonstrating this match visually - it is not self-evident from the curve plot provided.

We agree. We have revised it completely to make it clearer.

In Figure 3 it is also unclear how all three wood objects have the same ring numbering. This is explained in the methods section at the end of the paper (counting back from the waney edge), but it would be helpful were it made equally clear in the main text for the benefit of an interdisciplinary readership. Many will not be familiar with the conventions of dendrochronology.

This journal does not allow for long methodological explanations in the main text. But we agree with the reviewer that this would be helpful and so have added a sentence.

Given how the methodology is explained in the text (e.g. ascertaining the 993 CE ring in each sample by wiggle matching, and then counting forward to the felling dates of 1021 CE) the authors may wish to reconsider Figure 3b. It shows the results of the 14C chi-square test for 1021 CE, but does not show 993 CE. There is thus a mismatch between the thought process in the text and what is shown in the main figure. It may be clearer to readers if the authors illustrate the 14C attribution of the 993 CE rings (rather than the 1021 rings), and then also provide a drawing or photographs of the tree ring sequence(s) from the 993 CE ring(s) to the waney edge(s), either for all three timbers or one example. Certainly an image of one of the actual tree ring sequences is needed in the main text. At present they occur only in the extended figures.

As above, we have fixed Figure 3 and the accompanying explanation. Secondly, the Extended Data (c.f. the SI) is part of the downloadable copy of our article, so we think the images are already prominent enough. Lastly, our method does not involve the ring-width analysis of conventional dendrochronology and in fact, ring-width patterns are not relevant to our study, only the isotopic ratios they contain.

Slightly more could be made of the seasonality evidence, if positively evaluated by a method expert. Photographic evidence of this seasonality, based on early and late wood development, would then be valuably moved from the extended figures to the main text. If space is short to add these images, the present simple location map (Figure 1) could be reduced and made part of a multi-pane figure.

Two of our co-authors are indeed renowned dendrologists. We have taken the analysis of seasonality as far as we feel comfortable doing (summer 1021 CE).

Slightly more consideration is merited regarding the efficacy of the method used to remove PEG

contamination. The approach (soaking in hot water) is noted, but not how comprehensively it works, or how the authors would know if it did not.

We have added further sentences to the Methods. In short, we referred to previous studies which stated that water should be sufficient for its removal (Bruhn et al. 2001; Brock et al. 2018; Ensing et al. 2019). We also looked into the physical properties given by chemical manufacturers (e.g. Sigma Aldrich). On this basis, bearing in mind the nature of consolidants can change over the longer term, we applied the most rigorous treatment practicable (water, 80°C, 36 h). Nonetheless, in ¹⁴C pretreatment it is difficult to prove any contaminant has been completely eliminated. As the ¹⁴C/¹²C is infinitesimally small, conventional spectroscopic techniques (FTIR, UV-Vis, IRMS) are somewhat unsatisfactory. Thus, the best evidence comes from our quality control procedures. We conducted 21 full-process replicates, some split between two different laboratories, on samples of PEG-treated wood, of which 19 passed the χ^2 -text for equivalence at 95% probability (Extended Data Table 2). Such equivalence would be highly improbable if any residual contamination remained. Moreover, the results we obtained on the three sequences (two PEG-treated and one PEG-free) very closely follow a 40-year stretch of the annual IntCal20 reference curve. This congruence also suggests that contamination was highly unlikely to be distorting our results.

Referee #3 (Remarks to the Author):

By applying recent advances in radiocarbon dating methods to four wooden objects retrieved from a Norse settlement in L'Ance aux Meadows, Newfoundland, the authors have been able to date the felling of three trees to 1021 CE. The objects' origin in Norse activities is testified by traces of metal tools on them; the indigenous population did not have metal tools. The text is easily read, facts and argument are presented clearly and lucidly. The approach seems valid, and the quality of the archaeological data, which is what this reviewer has expert knowledge in, appears to be high. The resulting datings of the wooden samples seem reliable.

We appreciate the reviewer's endorsement of our analysis.

Hitherto, the dating of this settlement has relied on saga evidence, artefacts retrieved during excavations, and standard-type radiocarbon datings. While the latter two allow rather wide date ranges, the sagas indicate that periodic settlement in Vinland stretched over c. 3–13 years around 1000 AD. This is where our approach diverges from the reviewer's perspective. We never made use of the Icelandic legends in our attempts to achieve a date. There were several reasons for this. Firstly, our objective was to produce the first *scientific* date for the site, i.e. one that was independent of any textual interpretations. Secondly, there is considerable disagreement about the veracity of the Sagas, even in the historical and archaeological communities. Some researchers depend heavily on them whereas others disregard them entirely. Certainly, the stories were transmitted by word of mouth from generation to generation for centuries before they were finally written down. Thirdly, only two Sagas are relevant to our study and even they don't agree with each other (one has multiple trips and the other really only one) and they both contain patently non-historical, fantastical events, such as talking dead people. On the whole, it remains unclear how factual these stories really are (e.g., Halldórsson 2001; Cormack 2007; Barraclough 2016).

As carefully stated by the authors, the new dating concerns Norse presence, not the start or end of the Norse settlement. Bearing in mind that sagas do not necessarily mention all Norse journeys to Newfoundland, the site may have been periodically inhabited over a period of more than 13 years. Therefore, the bulk of activity may still lie a decade or two prior to the new dating. Therefore, the new dating does not significantly alter existing assessments of the Norse presence in Newfoundland, although one might now argue that it likely elapsed after the turn of the millennium.

This viewpoint is again based entirely on descriptions given in the Sagas, and reveals that the accuracy and comprehensiveness of these stories is disputable. Indeed, it is somewhat contradictory to say our date simply provides support for pre-existing historical interpretations, whilst also saying that these records omit the voyage needed to accommodate our date.

The Norse journeys to north-eastern America is a spectacular but minor episode in Viking History at large, and this reviewer finds that the authors overstate the significance of their findings somewhat.

One might be justified in asking whether it is really the case that “Our new date [...] sheds new light on Viking chronology” (lines 30–31)? In this reviewer’s estimate, the dating’s repercussions for Viking chronology at large are negligible. The authors are advised to limit the cited passage to ‘Viking Chronology on the American Continent’.

Whilst we accept that in the context of Viking history this episode may be somewhat peripheral, we are convinced that its significance to human history is actually difficult to overstate. In a nutshell, it represents the first time in which human contact spanned the globe. Indeed, some regard it as the beginning of the era of globalisation. Furthermore, it represents the first contact between the Old and New Worlds for some 10,000 years (after the formation of the Bering Strait).

Likewise, it seems hard to accept that the new dating “provides a definitive tie-point for future research into the consequences of the western expansion of Viking culture” (lines 31-2). The statement might convey the impression that the authors hold their dating to be significant for research into the consequences of Viking settlement in other western lands: Ireland, the British Isles, Normandy, Shetland, the Faroes, Iceland, and Greenland. If that is indeed what they mean, they need to present an argument for their case. Would, for instance, the two studies referred to on this issue (nos. 7–8) have arrived at different conclusions had their authors had access to the new dating? If this is not what they mean, the authors are advised to limit the scope of the cited passage to ‘the expansion of Viking culture to the American Continent’. In conclusion, this reviewer would advise that these and other (see also lines 164–5) rather far-reaching statements are left out altogether and that the authors instead turn their attention to discussing what their new dating has to say for the understanding of the Norse presence in north-eastern America in general and for the site in particular. For example, is it likely that the wooden items from 1021 were produced during one of the stays mentioned in the sagas, or do they more likely result from a subsequent and possibly unrecorded stay?

We do not pretend that our date rewrites the entire chronology of the Western expansion of the Vikings. We understand the confusion. By ‘western expansion’ we were really referring to the furthest western reaches (Greenland and Vinland) of Norse culture. In this context, we do think it is important, as there is so little first-hand evidence here that can be used for chronology building. However, in accordance with the reviewer’s comments we have adjusted the wording in the text to make this clearer. But, as we have stated above, the true significance of this milestone lies in what it marks for the peopling of the planet.

Referee #4 (Remarks to the Author):

A. This paper represents a very nice piece of work on the dating of the brief Norse occupation of N America using what is essentially a combination of radiocarbon and dendrochronology. It makes particular use of the rapid events in the radiocarbon record assumed to be due to unusual solar activity.

B. The paper is a novel application of this combination of techniques and gives new insight with a precision that is at least an order of magnitude better than previously attainable. C. The overall methods and approach look appropriate. The data is clearly of high quality both from the inter-comparisons but also from the internal consistency of the data set.

Again, we thank the reviewer for their positive appraisal of our study.

D. In general the use of statistics looks appropriate. However, there is one area which needs looking at in more detail. The Bayesian approach provides the initial analysis and is then backed up with a classical approach using a chi-squared test to find the best fit. This is generally a valid approach but you would expect the lowest chi-squared values to correspond to the highest Bayesian probabilities. I cannot see why sample 4A 68 J4-6 shown in blue in Fig 2c has its highest probability a couple of years earlier than the other samples and yet the same sample has the lowest chi-squared at the same year in Fig 3. This may just be a result of the way the uncertainties in the underlying curve fall relative to the data.

This is well observed. However, the wiggle matches in OxCal are not directly comparable with the χ^2 tests. This was not clear in our text, and we have since rectified the issue. In short, the wiggle-matches against the smoothed IntCal20 curve used all of our data whereas the χ^2 tests focused on a subset of samples (see our response to Reviewer 1’s comments about Line 126 and 140). Indeed, we have now chosen to use the χ^2 on another reference. However, we would like to thank the reviewer for this

comment, because when redoing the wiggle-matches we discovered an error of 0.5 years which we have also fixed (Fig 2c).

However, more generally I think it is important to argue more carefully for the single year hypothesis. Using either the Bayesian or the classical approaches a number of different fits would be possible - that which is presented is only the most likely. The fact that it also happens to give the same felling date to all the samples is extra support for this - but we probably cannot rule out a year either side of this. By assuming the samples are all the same age the Bayesian approach could give error limits on a combined felling date. Certainly the most likely would be that presented but we probably could not rule out adjacent years at the 95% confidence level.

This point is also addressed above in our response to Reviewer 1's comments about Line 126 and 140. In short, the former use all the data we obtained for each piece of wood to find the general range within which the felling occurred (1019–1024 CE, meaning the 993 CE anomaly must be between rings -31 and -26). The nature of the OxCal algorithm, and the fact we apply it against the smoothed IntCal20 curve mean a range is inevitable using this approach. But this belies the true situation, however, that there is a unmistakable drop in NH ¹⁴C data between 992–994 CE. So we take six-ring subsets (rings -31 to -26) and match them using the χ^2 test against B2018, a comprehensive annual reference data set published in 2018 which more precisely reveals the anomaly. This identifies ring -29 as year 992 CE and hence 1021 CE as the cutting year. But as also mentioned above, we now list the other dates that are also possible at 95% probability, albeit less likely.

What is perhaps more compelling as an argument is that fact that all three samples show the rapid drop in the same year with this conclusion.

Agreed. In fact, this is exactly what we were trying to do. We have revised Figure 3a and added a few sentences to make this point clearer. With B2018, we believe it is now much more obvious.

E. Despite the points above the overall results look very robust: it is very hard to see how this could be significantly wrong in terms of the main conclusion.

F. I would suggest some minor extra discussion on the statistical front. The other area which it would be good to see some discussion in the main text is the origin of the wood in the artefacts. From the arguments of the chronology, I assume the understanding is that these artefacts are of N American wood - made on site. This is never explicitly said and is an important element in the argument. Otherwise these could be material brought from Europe and potentially a few years before.

We discuss this in line 146-152.

G. The references seem appropriate

H. The paper is clearly written and the abstract adequately describes the content.

References:

- Barracough, E. R. *Beyond the Northlands: Viking Voyages and the Old Norse Sagas*. (Oxford, Oxford University Press, 2016).
- Brock, F., Dee, M. W., Hughes, A., Snoeck, C., Staff, R. & Bronk Ramsey, C. 2018. Testing the effectiveness of protocols for removal of common conservation treatments for radiocarbon dating on dating. *Radiocarbon* **60**(1): 35-50.
- Bruhn, F., Duhr, A., Grootes, P. M., Mintrop, A., Nadeau, M-J. 2001 Chemical removal of conservation substances by 'Soxhlet'-type extraction. *Radiocarbon* **43**: 229–237.
- Cormack, M. Fact and Fiction in the Icelandic Sagas. *History Compass* **5** (2007), 201-207.
- Gleeson, P. Study of Wood Material from L'Anse aux Meadows. *Unpublished report for Parks Canada Atlantic Service Centre*, **3**, 9-13 (Halifax, 1979).
- Ensing, B., Tiwari, A., Tros, M., Hunger, J., Domingos, S. R., Pérez, C., Smits, G., Bonn, M., Bonn, D. & Woutersen, S. 2019. On the origin of the extremely different solubilities of polyethers. *Nature Communications* **10**: 2893.
- Halldórsson, Ó. "The Vinland Sagas," in A. Wawn and Þórunn Sigurðardóttir (eds.), *Approaches to Vinland* (Reykjavík: Sigurður Nordal Institute, 2001).
- Sigma Aldrich Product Information (polyethylene Glycol, PEG):

www.sigmaaldrich.com/deepweb/assets/sigmaaldrich/product/documents/215/480/p4463pis.pdf

Wallace, B. L. *Westward to Vinland, The Saga of L'Anse aux Meadows*. (St. John's: Historic Sites Association of Newfoundland and Labrador, 2012).

Reviewer Reports on the First Revision:

Referee #1 (Remarks to the Author):

Dear authors,

I congratulate the authors for the very nice work they present in their manuscript! I strongly recommend a publication. The authors propose the first accurate and precise date for early European presence (by Vikings) in North America. This is a most significant finding which verifies and supports for the first time precisely the dates given by the famous 13–15th century Icelandic Sagas. The work proves an early settlement in L'Anse aux Meadows in 1021 AD, far earlier than the discovery of America by Columbus, what is certainly of highest interest to a broad audience. The researchers applied state-of-the-art methods using high precision radiocarbon dating. While the method using radiocarbon events for precise and accurate dating is not applied for the first time, it is certainly the most significant date produced so far in Archeology using this new method. The results are highly significant and securely verified (several objects with replications by 2 laboratories).

The statistical approach applied is after revision now correct, conclusive and clear. It is correct, but is not always conclusive and clear (see also comments).

The message is now clear and the manuscript reads well. References are appropriately given. No further edits are required in my opinion. Well done!

Referee #2 (Remarks to the Author):

The revised manuscript successfully addresses almost all issues raised in the initial peer review. The suggestion that the surfaces cut by metal tools be shown in photographs has not been adopted. Nor has the recommendation that an image of the ring-growth patterns (perhaps regarding the seasonality attribution) be moved into the main paper. I remain of the opinion that the paper would benefit from these additional changes; the attribution to metal tools is otherwise established by an unpublished report, and the interdisciplinary essence of the paper is lost by excluding the dendrochronology images from the main text. Nevertheless, the paper is certainly now publishable (after correcting a minor typo on line 151: "We provides evidence") and will be a valuable contribution to knowledge.

Referee #3 (Remarks to the Author):

Following comments from reviewers, the authors have made significant adjustments regarding the framing of the new dating from L'Anse aux Meadows. The shift from "Our new date [...] sheds new light on Viking chronology" to "Our new date represents the first point at which humans encircled the globe" (line 31, repeated in lines 153–4) is a real shift in perspective from Vikings to the global history of the human species.

However, for two reasons, the new framing is not accurate:

1. The 1021 dating is evidence that Norse individuals were present in L'Anse aux Meadows in that year. However, it does not date the first encounter between the Norse and the indigenous population in Newfoundland. This first encounter might well have taken place several decades prior to 1021, or perhaps later.
2. More importantly, when the Norse settled Greenland from the mid-980s onwards, they will soon have encountered the indigenous population, the so-called 'Dorset culture', a Paleo-Eskimo culture whose ancestors arrived at the island from the Americas. The indigenous population's first encounter with the Norse in Greenland was probably 'the first point at which humans encircled the

globe'. While the Norse settlement in Newfoundland was tiny, sporadic, and short-lived, that in Greenland consisted of hundreds of peoples and lasted into the 15th century. Thus, the chances for "the transference of knowledge, and the potential exchange of genetic information, biota and pathologies" (lines 33-34), if it at all happened, would be much greater in Greenland. Recent evidence indicating that genetic information was not exchanged in Greenland (note 8) renders the suggestion that it may have happened in Newfoundland somewhat speculative.

In their rebuttal document, the authors claim that their "approach diverges from the reviewer's perspective" in that they "never made use of the Icelandic legends in our attempts to achieve a date". Of course, I never claimed that they did; the 1021 dating might have been obtained whether or not the sagas existed. What I claim is that the saga evidence contributes to the context within which the authors situate the new dating. They repeatedly refer to saga evidence in the new version (lines 47, 52, 142, Extended Data Fig.3, and Supplementary information sections 1.2 and 3.2). Actually, the core element of their present framing of the dating, contact between the Norse and the indigenous population, depends on the sagas: "The literary and archaeological records show that the Norse engaged in cultural exchanges with the indigenous groups of North America" (lines 142-3). While the sagas refer to hostilities with the indigenous population as the main reason for abandoning the settlement in the Americas, the archaeological evidence for this contact is utterly weak. The few Norse items found in indigenous sites might have been obtained from a Norse site after it was abandoned.

The saga evidence is far from perfect for dating purposes, and surely, they mention only some of the settlement episodes in the Americas. Therefore, I applaud the authors' efforts to produce a scientific dating from L'Anse aux Meadows. However, the start and the end dates of Norse presence in L'Anse aux Meadows are still unknown, and the 1021 dating falls roughly within the range of existing assessment based in saga evidence. Therefore, I maintain that the contribution presents new knowledge, but not of a significant nature for the dating and understanding of Norse settlement in Newfoundland.

In conclusion, I will repeat my advice to abstain from sweeping statements (cited in the first paragraph above) that have, at best, a frail basis in the evidence, and instead frame the new dating within the historical context of the site: the Norse presence in north-eastern America in general and in this site in particular.

Referee #4 (Remarks to the Author):

I think the authors have addressed the suggestions in the review sufficiently and that the paper is now ready for publication.

Author Rebuttals to First Revision:

Referee #1 (Remarks to the Author):

Dear authors,

I congratulate the authors for the very nice work they present in their manuscript! I strongly recommend a publication. The authors proposes the first accurate and precise date for early European presence (by Vikings) in North America. This is a most significant finding which verifies and supports for the first time precisely the dates given by the famous 13–15th century Icelandic Sagas. The work proves an early settlement in L'Anse aux Meadows in 1021 AD, far earlier than the discovery of America by Columbus, what is certainly of highest interest to a broad audience.

The researchers applied state-of-the art methods using high precision radiocarbon dating. While the method using radiocarbon events for precise and accurate dating is not applied for the first time, it is certainly the most significant date produced so far in Archeology using this new method. The results are highly significantly and securely verified (several objects with replications by 2 laboratories).

The statistical approach applied is after revision now correct, conclusive and clear. are in correct, but is not always conclusive and clear (see also comments).

The message is now clear and the manuscript reads well. References are appropriately given.

No further edits are required in my opinion. Well done!

We thank the referee for the enthusiastic comments. We would also like to express our gratitude for the expert advice - the suggested changes have certainly improved the paper.

Referee #2 (Remarks to the Author):

The revised manuscript successfully addresses almost all issues raised in the initial peer review. The suggestion that the surfaces cut by metal tools be shown in photographs has not been adopted. Nor has the recommendation that an image of the ring-growth patterns (perhaps regarding the seasonality attribution) be moved into the main paper. I remain of the opinion that the paper would benefit from these additional changes; the attribution to metal tools is otherwise established by an unpublished report, and the interdisciplinary essence of the paper is lost by excluding the dendrochronology images from the main text. Nevertheless, the paper is certainly now publishable (after correcting a minor typo on line 151: "We provides evidence") and will be a valuable contribution to knowledge.

We thank the referee for both rounds of feedback. We had regarded the work on the cut surfaces to be *a priori* knowledge, upon which we were building. However, in light of this referee's comments, we have now decided to include more information from the earlier studies. Specifically, we have added more detail to our Supplementary Information Note 4. Here, we refer to item 4A 68 E2-2 which is described by Gleeson (1979) as depicting the 'distinctive shearing and well-defined ridging of a metal flat, low angle blade edge'. We would also like to reiterate that these matters were addressed by Wallace (2012); and that the Gleeson (1979) report is available upon request from Parks Canada. In terms of the other dendrochronological aspects of our study, we do include images of the ring-growth patterns, but strict constraints on space mean we are unable to move these into the main text.

Referee #3 (Remarks to the Author):

Following comments from reviewers, the authors have made significant adjustments regarding the framing of the new dating from L'Anse aux Meadows. The shift from "Our new date [...] sheds new light on Viking chronology" to "Our new date represents the first point at which humans encircled the globe" (line 31, repeated in lines 153–4) is a real shift in perspective from Vikings to the global history of the human species.

However, for two reasons, the new framing is not accurate:

1. The 1021 dating is evidence that Norse individuals were present in L'Anse aux Meadows in that year. However, it does not date the first encounter between the Norse and the indigenous population in Newfoundland. This first encounter might well have taken place several decades prior to 1021, or perhaps later.

We agree and have added some details to the text to make this clearer. Indeed, we made sure throughout not to describe it as the first, last or middle year of Norse activity. All three of our samples return 1021 CE. Our contention is really that this is the only calendar year, and therefore fundamental time-marker, in which Pre-Columbian presence by Europeans in the Americas can be scientifically demonstrated. The question of when or if contact was made with the local population is even more complex (see below).

2. More importantly, when the Norse settled Greenland from the mid-980s onwards, they will soon have encountered the indigenous population, the so-called 'Dorset culture', a Paleo-Eskimo culture whose ancestors arrived at the island from the Americas. The indigenous population's first encounter with the Norse in Greenland was probably 'the first point at which humans encircled the globe'. While the Norse settlement in Newfoundland was tiny, sporadic, and short-lived, that in Greenland consisted of hundreds of peoples and lasted into the 15th century. Thus, the chances for "the transference of knowledge, and the potential exchange of genetic information, biota and pathologies" (lines 33-34), if it at all happened, would be much greater in Greenland. Recent evidence indicating that genetic information was not exchanged in Greenland (note 8) renders the suggestion that it may have happened in Newfoundland somewhat speculative.

We believe that much of this criticism is not supported by the evidence. There is no scientific, archaeological, or even Saga evidence the Dorset Culture were present in the vicinity of the Norse Greenlandic settlements in the few decades between their founding and the arrival of the Norse at LAM (Arneborg 2003; Campbell 2021). In fact, Arneborg (2003), Park (2008) and Smith *et al.* (2018) conclude that no evidence exists anywhere for Dorset-Norse contact. The Sagas mention the settlers found artefacts indicating that Greenland had once been populated (Arneborg 2003), but no indigenous presence is ever mentioned. By contrast, material and written evidence does exist for encounters with the Thulé on Greenland; however, these all began centuries after 1021 CE (Scheldermann & McCullough 2003; Arneborg 2003; Ramsden & Rankin 2013).

In their rebuttal document, the authors claim that their "approach diverges from the reviewer's perspective" in that they "never made use of the Icelandic legends in our attempts to achieve a date". Of course, I never claimed that they did; the 1021 dating might have been obtained whether or not the sagas existed. What I claim is that the saga evidence contributes to the context within which the authors situate the new dating. They repeatedly refer to saga evidence in the new version (lines 47, 52, 142, Extended Data Fig.3, and Supplementary information sections 1.2 and 3.2). Actually, the core element of their present framing of the dating, contact between the Norse and the indigenous population, depends on the sagas: "The literary and archaeological records show that the Norse engaged in cultural exchanges with the indigenous groups of North America" (lines 142-3). While the sagas refer to hostilities with the indigenous population as the main reason for abandoning the

settlement in the Americas, the archaeological evidence for this contact is utterly weak. The few Norse items found in indigenous sites might have been obtained from a Norse site after it was abandoned.

Here a clear distinction can be made: the basis for the date, and the likely circumstances the Norse encountered. The saga evidence is unquestionably part of the historical context, and precious tradition of human travels and exploration. However, this context did not affect the establishment of our date at all. This result of 1021 CE is wholly independent of the Sagas. It relies only on metal-modified wood, isotope measurements, and a cosmic radiation event in 992 CE. Once this fact is established, we do not see any contradiction in then citing additional archaeological, environmental and oral-history records about the situation the Norse are likely to have faced. Estimates of the indigenous population of the Canadian Maritimes at the time number in the tens of thousands (Dickason 1992). The Sagas claim there were interactions between the locals and the Norse. However, as this referee points out, little archaeological evidence has been found to substantiate such interactions. Our study just acts as a departure point for further empirical work on this matter, as well as any possible ecological footprint left by the Europeans. If no such evidence is ever found, and the Sagas cast into more doubt, our date will still stand, with its importance potentially enhanced.

The saga evidence is far from perfect for dating purposes, and surely, they mention only some of the settlement episodes in the Americas.

We would argue that multiple trips cannot yet be scientifically substantiated. It remains possible that they only went once.

Therefore, I applaud the authors' efforts to produce a scientific dating from L'Anse aux Meadows. However, the start and the end dates of Norse presence in L'Anse aux Meadows are still unknown, and the 1021 dating falls roughly within the range of existing assessment based in saga evidence. Therefore, I maintain that the contribution presents new knowledge, but not of a significant nature for the dating and understanding of Norse settlement in Newfoundland. In conclusion, I will repeat my advice to abstain from sweeping statements (cited in the first paragraph above) that have, at best, a frail basis in the evidence, and instead frame the new dating within the historical context of the site: the Norse presence in north-eastern America in general and in this site in particular.

Actually, 1021 CE does not sit comfortably with most readings of the Sagas, and we expect scholars of these records to be intrigued by this result. However, we do not contend that it dramatically alters understanding of LAM, or indeed the history of the Vikings. That is not the crux of our work. We use a brand new technique to prove that Europeans were active in the Americas in exactly 1021 CE. Discussions around whether they were there earlier or later, for long periods, or only once, are at present heavily dependent on the oral histories. However, it is now **empirically demonstrable** that from exactly 1021 CE onward, Europeans knew of the Americas, and that by this year the Atlantic Ocean had been crossed, and human migration had finally encircled the entire planet. We think this is far more profound than the relevance of the date for this archaeological site.

Referee #4 (Remarks to the Author):

I think the authors have addressed the suggestions in the review sufficiently and that the paper is now ready for publication.

We thank the referee for this supportive review. We would also like to reiterate how perceptive this referee's original comments were, and how addressing them has resulted in significant improvements to our manuscript.

Arneborg J. Norse Greenland: reflections on settlement and depopulation. In: *Contact, Continuity, and Collapse: the Norse colonization of the North Atlantic*. J Barrett (ed). Turnhout: Brepols. 163-182 (2003).

Campbell G. *Norse America The Story of a Founding Myth*. Oxford: Oxford University Press (2021).

Dickason, O. P. *Canada's First Nations: A History of Founding Peoples from Earliest Times*. Oxford: Oxford University Press (1992).

Gleeson, P. Study of Wood Material from L'Anse aux Meadows. *Unpublished report for Parks Canada Atlantic Service Centre*, **3**, 9-13 (Halifax, 1979).

Park R. W. Contact between the Norse Vikings and the Dorset culture in Arctic Canada. *Antiquity* **82**: 189–198 (2008)

Schledermann P. & McCullough K. M. Inuit-Norse contact in the Smith Sound region. In: *Contact, continuity, and Collapse: the Norse colonization of the North Atlantic*. ed. J Barrett. Turnhout: Brepols. 183-206 (2003).

Ramsden P. & Rankin L. K. Thule radiocarbon chronology and its implications for early Inuit–European interaction in Labrador. In: *Exploring Atlantic transitions: archaeologies of transience and permanence in new found lands*. Pope P. E. & Lewis-Simpson S. (eds). Woodbridge: The Boydell Press. 299-309 (2013).

Smith M. H., Smith K. P., Nielson G. Dorset, Norse, or Thule? Technological transfers, marine mammal contamination, and AMS dating of spun yarn and textiles from the Eastern Canadian Arctic. *Journal of Archaeological Science* **96**: 162–174 (2018).

Wallace, B. L. *Westward to Vinland, The Saga of L'Anse aux Meadows*. St. John's: Historic Sites Association of Newfoundland and Labrador (2012).